# Targeted and high-throughput gene knockdown in diverse bacteria using synthetic sRNAs

Jae Sung Cho [1,2,5,8], Dongsoo Yang [1,2,6,8], Cindy Pricilia Surya Prabowo [1,2], Mohammad Rifqi Ghiffary [2,3], Taehee Han [1,2], Kyeong Rok Choi [1,2], Cheon Woo Moon [1,2], Hengrui Zhou[1,2], Jae Yong Ryu[1,2,7], Hyun Uk Kim [2,3,4] & Sang Yup Lee [1,2,4] ✉

Synthetic sRNAs allow knockdown of target genes at translational level, but have been restricted to a limited number of bacteria. Here, we report the development of a broad-host-range synthetic sRNA (BHR-sRNA) platform employing the RoxS scaffold and the Hfq chaperone from *Bacillus subtilis*. BHR-sRNA is tested in 16 bacterial species including commensal, probiotic, pathogenic, and industrial bacteria, with >50% of target gene knockdown achieved in 12 bacterial species. For medical applications, virulence factors in *Staphylococcus epidermidis* and *Klebsiella pneumoniae* are knocked down to mitigate their virulence-associated phenotypes. For metabolic engineering applications, high performance *Corynebacterium glutamicum* strains capable of producing valerolactam (bulk chemical) and methyl anthranilate (fine chemical) are developed by combinatorial knockdown of target genes. A genome-scale sRNA library covering 2959 *C. glutamicum* genes is constructed for high-throughput colorimetric screening of indigoidine (natural colorant) overproducers. The BHR-sRNA platform will expedite engineering of diverse bacteria of both industrial and medical interest.

Advances in synthetic biology and metabolic engineering have enabled the efficient engineering of model bacteria for both biomedical[1,2] and industrial[3,4] applications. In the medical applications, a number of model probiotic bacteria (e.g., *Escherichia coli* Nissle 1917) have been engineered to carry out therapeutic[1,5] or diagnostic[6] tasks, but the vast majority of bacteria including human pathogens and commensal bacteria still lack tools for the systematic interrogation of gene expression and synthetic biology-based cell therapies. In the industrial

biotechnology applications, the construction of efficient microbial cell factories to produce chemicals and materials from renewable resources relies on fine-tuning metabolic and regulatory networks[4], but the identification and experimental validation of potential gene targets by chromosomal manipulation are labor-intensive and time-consuming. The use of *trans*-acting target gene knockdown systems such as CRISPR interference (CRISPRi) allows rapid knockdown of target genes at the transcriptional level without chromosomal manipulation[7], and

[1]Metabolic and Biomolecular Engineering National Research Laboratory, Department of Chemical and Biomolecular Engineering (BK21 four), Institute for the BioCentury, Korea Advanced Institute of Science and Technology (KAIST), Daejeon 34141, Republic of Korea. [2]Systems Metabolic Engineering and Systems Healthcare Cross-Generation Collaborative Laboratory, KAIST, Daejeon 34141, Republic of Korea. [3]Systems Biology and Medicine Laboratory, Department of Chemical and Biomolecular Engineering (BK21 four), KAIST, Daejeon 34141, Republic of Korea. [4]KAIST Institute for Artificial Intelligence, BioProcess Engineering Research Center and BioInformatics Research Center, KAIST, Daejeon 34141, Republic of Korea. [5]Present address: Department of Biological Engineering, Massachusetts Institute of Technology, Cambridge, MA, USA. [6]Present address: Department of Chemical and Biological Engineering, Korea University, Seoul 02481, Republic of Korea. [7]Present address: Department of Biotechnology, College of Science and Technology, Duksung Women's University, Seoul, Republic of Korea. [8]These authors contributed equally: Jae Sung Cho, Dongsoo Yang. ✉e-mail: leesy@kaist.ac.kr

the recent Mobile-CRISPRi[8] system can be used to knock down target genes in diverse bacteria. However, the practical applications of CRISPR-based tools in bacteria are sometimes limited due to the metabolic burden caused by the Cas9 protein[9].

Small RNAs (sRNAs) are short non-coding RNAs that control gene expression in bacteria at the translational level (Fig. 1a). They play important physiological roles including metabolic regulation, quorum sensing, and virulence[10]. Leveraging their natural function to regulate translation, we previously developed a systematic method to knock-down specific target genes in *E. coli* using reprogrammed synthetic sRNAs[11]. Binding of the synthetic sRNA to its target mRNA is facilitated

by the sRNA scaffold in the form of 3' hairpin structure and the Hfq chaperone[12] (Fig. 1b). Since then, the synthetic sRNA technology has been actively employed for engineering a handful of Gram-negative bacteria[11,13–16] and Gram-positive bacteria[17–19]. However, unlike the well-documented sRNA-mediated riboregulation in Gram-negative bacteria, the mechanism behind sRNA-mediated riboregulation in Gram-positive bacteria is still disputed and unclear[20]. Since the synthetic sRNA-based knockdown systems developed so far have been all solely based on sRNA scaffold and Hfq from *E. coli*[11] (Supplementary Table 1), there is a limitation in generally adapting the current sRNA knockdown tool for its use in diverse bacteria.

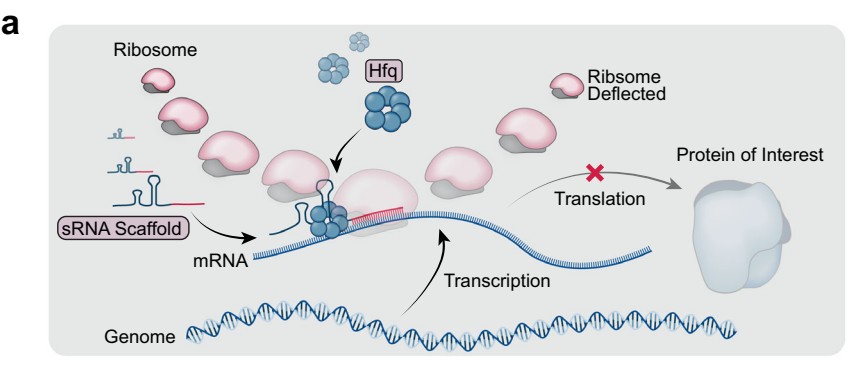

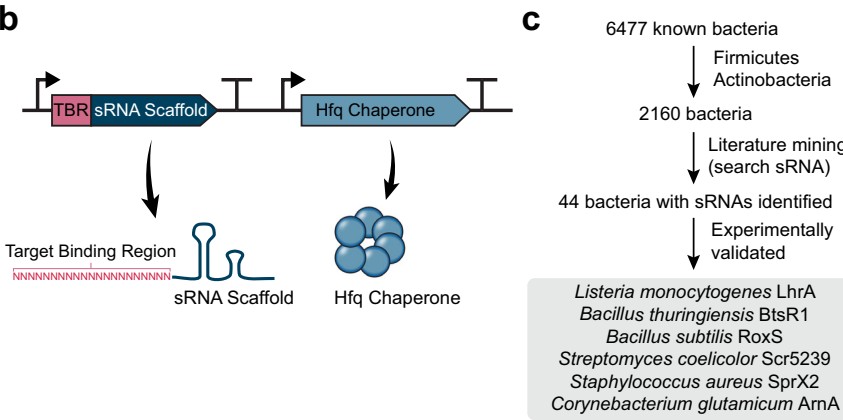

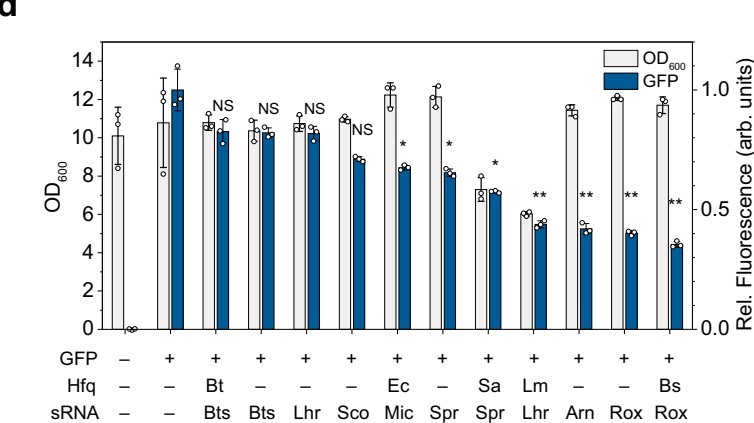

**Fig. 1 | Development of synthetic sRNA systems for the knockdown of target genes. a** Mechanism of synthetic sRNA-mediated translation repression. The sRNA scaffold and Hfq form a ribonucleoprotein complex and bind to a target binding region on the mRNA, resulting in the inhibition of protein translation by the ribosome. **b** Components for synthetic sRNA-based gene knockdown. TBR, target binding region. **c** Procedure taken to select sRNA candidates derived from Gram-positive bacteria. **d** Effects of various synthetic sRNA scaffolds and Hfq proteins on GFP fluorescence knockdown efficiency and cell growth of *C. glutamicum*.

Abbreviations for Hfq: Bs, *B. subtilis* Hfq; Ec, *E. coli* Hfq; Lm, *L. monocytogenes* Hfq; Sa, *S. aureus* Hfq. Abbreviations for sRNA scaffolds: Arn, ArnA; Bts, BtsR1; Lhr, LhrA; Mic, MicC; Sco, Scr5239; Spr, SprX1; Rox, RoxS. *$P < 0.0045$, **$P < 0.00091$, determined by two-tailed Student's *t*-test. *P*-value thresholds were adjusted using Bonferroni correction (corrected significance levels represented as $\alpha/m$; $\alpha$, original significance level; $m$, number of hypotheses). Error bar represents the mean ± standard deviation (SD; $n = 3$ biologically independent samples). NS, Not significant; arb. units, arbitrary units. Source data are provided as a Source Data file.

Here we report the development of a broad-host-range sRNA platform comprising sRNA scaffold and Hfq from *Bacillus subtilis*, with its versatility as a gene knockdown tool in diverse bacteria. As a medical application of this sRNA system, the virulence phenotypes are removed from pathogenic bacteria. Also, the sRNA system is applied to the metabolic engineering of different bacteria for the production of chemicals.

## Results

### Development of efficient sRNA-based gene knockdown systems

To test the performance of different sRNA systems, *Corynebacterium glutamicum* ATCC 13032, a widely used industrial bacterium[21], was first chosen for its taxonomic position as a Gram-positive bacterium and for its well-established engineering tools and protocols. We initially examined the previously developed *E. coli* MicC scaffold and *E. coli* Hfq (EcHfq)-based sRNA system[11] in *C. glutamicum*. While the MicC-EcHfq combination successfully knocked down green fluorescent protein (GFP) fluorescence in *C. glutamicum* (Supplementary Fig. 1a) as previously reported[19], the maximum knockdown efficiency was unexpectedly low (up to only 35%; $P = 5.2 \times 10^{-6}$; Supplementary Fig. 1a, b, Supplementary Note 1).

It was reasoned that 35% repression is insufficient for extensive engineering applications, even when the high expression level of the reporter gene from a multicopy plasmid is considered. As the sRNA-mediated gene regulation mechanisms in Gram-positive bacteria are still being debated[20], we postulated that the sRNA system from Gram-negative bacteria (i.e., MicC-EcHfq) is not suitable for wide use in diverse bacterial species including Gram-positive bacteria. To address this issue, we sought to screen all sRNA systems identified in Gram-positive bacteria that are potentially applicable for the knockdown of target genes. Among 2160 Gram-positive bacteria (Firmicutes and Actinobacteria) catalogued in the KEGG database, we found through literature mining that sRNAs have been reported in 44 bacteria (Supplementary Data 1). Of these sRNAs reported, six candidate sRNAs that had experimentally validated target binding sequences and scaffold sequences were selected: LhrA from *Listeria monocytogenes*[22], BtsR1 from *Bacillus thuringiensis*[23], RoxS from *Bacillus subtilis*[24], Scr5239 from *Streptomyces coelicolor*[25], SprX2 from *Staphylococcus aureus*[26,27], and ArnA from *C. glutamicum*[28,29] (Fig. 1b, c, Supplementary Table 2). With the exception of *S. coelicolor* and *C. glutamicum* for which no known Hfq proteins have been reported, all corresponding Hfq proteins were also introduced together with the corresponding sRNAs for examining their knockdown efficiencies in vivo (Supplementary Table 3). The corresponding scaffold sequences were taken from their native sRNA sequences and the target binding sequence of each native sRNA was replaced with the antisense sequence to the first 24 nucleotides (nt) of the *GFP* gene (Fig. 1b). Among the candidate sRNA systems, the RoxS and *B. subtilis* Hfq sRNA system (RoxS-BsHfq) encoded in the plasmid pEK-bhr showed up to 64.6% knockdown ($P = 5.2 \times 10^{-3}$; Fig. 1d, Supplementary Fig. 1c) with an optimal target binding sequence length of 24 nt (Supplementary Fig. 1d). The scaffold RoxS only also resulted in 58.0% knockdown of *GFP*, but the knockdown efficiency was lower than that (64.6%) by the RoxS-BsHfq system ($P = 0.012$). Thus, the RoxS-BsHfq system was designated as a potential broad-host-range synthetic sRNA system (BHR-sRNA system) for further testing.

Next, the BHR-sRNA system was tested for the simultaneous knockdown of multiple target genes. Employing sfGFP and mCherry as the dual fluorescent reporters, simultaneous knockdown of the both reporters was achieved by the introduction of a single plasmid harboring the anti-*sfGFP* and the anti-*mCherry* sRNAs (Supplementary Fig. 1e). Orthogonal knockdown of either reporter was also achieved by the introduction of each sRNA to the *C. glutamicum* strain harboring the dual reporters. The maximum cell growth (represented as OD$_{600}$) was not affected by the introduction of the BHR-sRNA system ($P = 0.54$;

Fig. 1d), which is advantageous over the CRISPRi system that impeded cell growth (Supplementary Fig. 2a) likely due to the increased metabolic burden by the large-size dCas9 protein as reported in *E. coli*[30] (Supplementary Note 2). We also observed that neither BsHfq nor the BHR-sRNA system influenced the growth profile of *C. glutamicum* when grown in a chemically defined CGXII medium (Supplementary Fig. 2b). Furthermore, quantitative reverse transcription PCR (RT-qPCR) showed that the GFP transcript levels were not affected by the introduction of the RoxS system ($P = 0.92$) and the RoxS-BsHfq system ($P = 0.91$), confirming that the target gene knockdown by the BHR-sRNA system was indeed due to translation-level repression (Supplementary Fig. 2c). In addition, we tested whether employing the BHR-sRNA system together with CRISPRi can further enhance knockdown efficiencies by the dual repression at both transcriptional and translational levels. Although the knockdown efficiency by the dual BHR-sRNA/CRISPRi system (78.5%) was higher than that (64.6%) by the BHR-sRNA system (Supplementary Fig. 2d), the dual BHR-sRNA/CRISPRi system was not chosen due to the decreased cell growth and the complexity of the system (Supplementary Fig. 2e). Comparison of the BHR-sRNA system with CRISPRi and the MicC-EcHfq system is summarized in Supplementary Table 4.

### The BHR-sRNA system is generally applicable in diverse bacteria

The BHR-sRNA-based knockdown system was first tested in *E. coli*, a representative and the best-studied Gram-negative bacterium, using the *EGFP* gene encoding enhanced green fluorescent protein (EGFP) as a representative target. Efficient knockdown of *EGFP* (83.9%; $P = 2.6 \times 10^{-8}$) in *E. coli* DH5α was confirmed (Fig. 2b). Then, the BHR-sRNA system was tested for its application in a broad range of Gram-negative and Gram-positive bacteria (Fig. 2a, b). Seven Gram-positive bacterial species (in addition to *C. glutamicum* ATCC 13032) corresponding to the phyla of Firmicutes (*Lactococcus lactis* subsp. *cremoris* MG1363, *Staphylococcus epidermidis* KCTC 13172, and *B. subtilis* subsp. *subtilis* strain 168) or Actinobacteria (*Rhodococcus opacus* PD630 DSM 43205, *Corynebacterium xerosis* ATCC 373, *C. glutamicum* BE, and *S. coelicolor* A3(2) ATCC 10147), and seven Gram-negative bacterial species (in addition to *E. coli* DH5α) corresponding to the classes of Betaproteobacteria (*Cupriavidus necator* H16 ATCC 17699 and *Chromobacterium violaceum* ATCC 12472) and Gammaproteobacteria (*Pseudomonas putida* KT2440 ATCC 47054, *Vibrio natriegens* ATCC 14048, *Aeromonas hydrophila* 4AK4, *Klebsiella pneumoniae* subsp. *pneumoniae* ATCC 15380, and *E. coli* Nissle 1917) were selected for testing. It should be noted that these selected bacterial species are either important to human health (commensal, probiotic, or pathogenic bacteria) or useful for the industrial production of chemicals and materials (Fig. 2a).

In these 14 different bacteria, the knockdown efficiency of BHR-sRNA system was tested using appropriate reporters. Due to the varying levels of difficulty in genetically manipulating these bacteria, three different strategies were employed. For *S. epidermidis*, *R. opacus*, *C. xerosis*, *C. glutamicum* BE, *C. necator*, *V. natriegens*, *A. hydrophila*, *K. pneumoniae*, and *E. coli* Nissle 1917, plasmids harboring genes encoding appropriate reporters (mRFPmars, EGFP, or GFP) were introduced to each strain by electroporation or conjugation (see Methods for details). For *B. subtilis* and *P. putida*, the *EGFP* gene was integrated into the respective chromosomes. For strains (*L. lactis*, *S. coelicolor*, and *C. violaceum*) where employing the two-plasmid system or chromosomal integration was difficult, knockdown of endogenous target genes that would result in phenotypic alterations was tested. In *L. lactis*, the *upp* gene (encoding uracil phosphoribosyltransferase) was selected as the knockdown target to examine the restoration of growth in the presence of toxic 5-fluorouracil[31]. *S. coelicolor* is known for its ability to produce the blue pigment actinorhodin, where knockdown of *actIORFI* encoding the ketosynthase of the minimal polyketide synthase would lead to the reduced production of actinorhodin[32]. *C. violaceum*

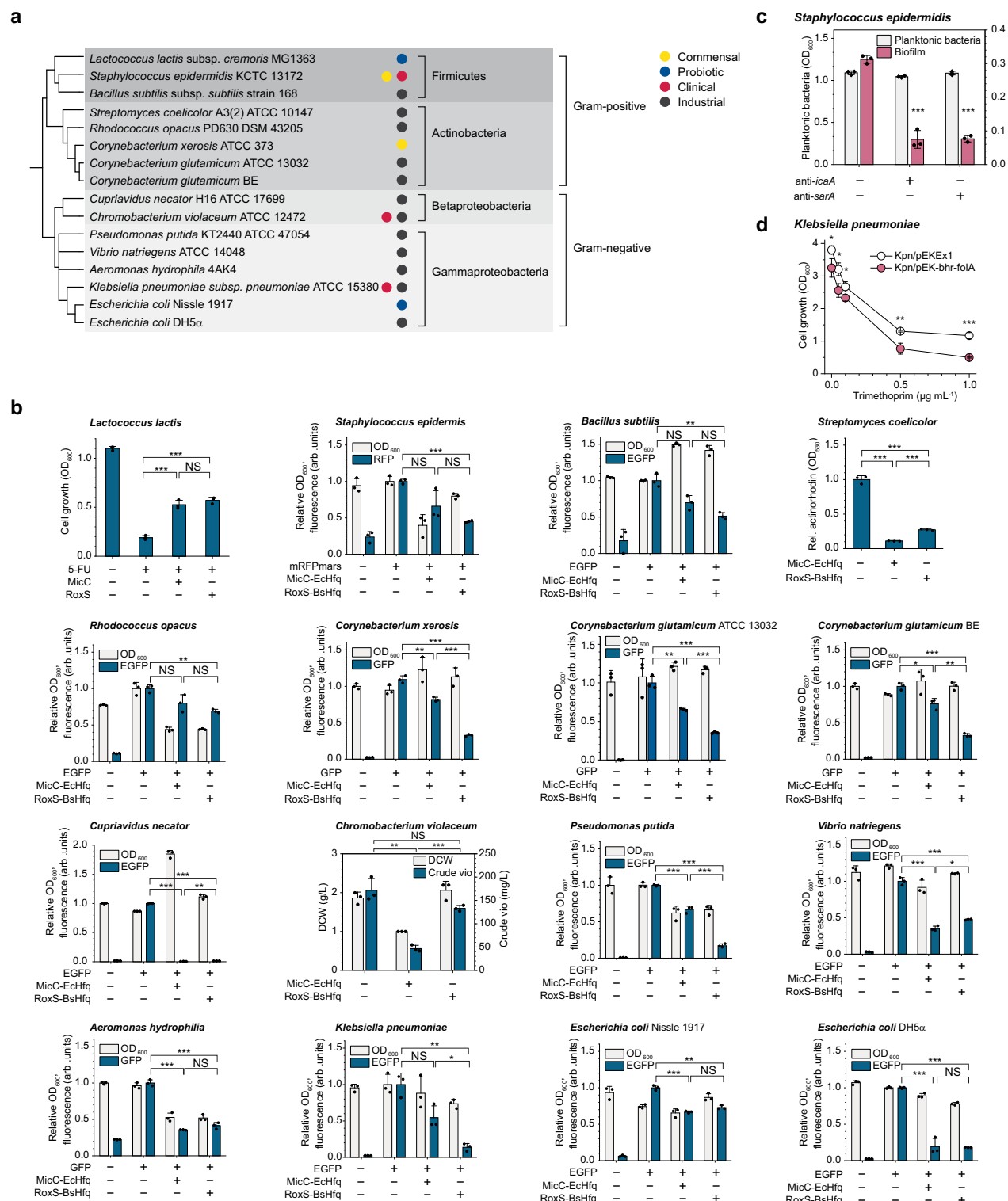

**Fig. 2 | Knockdown of target genes in different Gram-negative and Gram-positive bacteria. a** Phylogenetic tree of the Gram-negative and Gram-positive bacterial species tested for gene knockdown by the BHR-sRNA system. The phylogenetic tree was generated using phyloT (https://phylot.biobyte.de/) and visualized with iTOL[54]. **b** Knockdown of target genes related to distinctive phenotypes or genes encoding fluorescence reporters EGFP, GFP, or mRFPmars in various bacteria. The graph corresponding to *C. glutamicum* ATCC 13032 was rearranged from the data in Fig. 1d. Crude vio, crude violacein (violacein and deoxyviolacein); DCW, dry cell weight; *$P < 0.017$, **$P < 0.0033$, ***$P < 0.00033$, determined by two-tailed

Student's *t*-test. *P*-value thresholds were adjusted using Bonferroni correction ($P < \alpha/m$). NS, not significant; arb. units, arbitrary units. **c** Quantification of biofilm formed by *S. epidermidis* with or without sRNA (anti-*icaA* or anti-*sarA*), using the crystal violet assay. **d** Trimethoprim sensitivity of *K. pneumoniae* with or without anti-*folA* sRNA. Kpn, *K. pneumoniae*. (**c**, **d**) *$P < 0.05$, **$P < 0.01$, ***$P < 0.005$, determined by two-tailed Student's *t*-test. Error bars, mean ± SD ($n = 3$ biologically independent samples; in panel **d**, $n = 4$ biologically independent samples). Source data are provided as a Source Data file.

produces bluish purple dyes violacein and deoxyviolacein, so knockdown of the first gene *vioA* in the violacein biosynthetic operon *vioABCDE* was tested[33].

The BHR-sRNA system allowed successful knockdown of the reporter genes tested (Supplementary Table 5) in 15 out of 16 bacteria tested, with >50% of target gene repression achieved in six out of eight Gram-positive bacteria (*S. epidermidis*, *B. subtilis*, *S. coelicolor*, *C. xerosis*, *C. glutamicum* BE, and *C. glutamicum* ATCC 13032) and six out of eight Gram-negative bacteria (*C. necator*, *P. putida*, *V. natriegens*, *A. hydrophila*, *K. pneumoniae*, and *E. coli* DH5α), demonstrating the broad applicability of the BHR-sRNA knockdown system in a wide range of bacteria (Fig. 2b, Supplementary Fig. 3a). For testing in *L. lactis*, only RoxS was used to knockdown the *upp* gene as the construction of the sRNA plasmid harboring *BsHfq* was unsuccessful.

We also tested the knockdown of the above reporter genes in 16 bacteria by employing the MicC-EcHfq system, and found that the BHR-sRNA system outperformed the MicC-EcHfq system in seven out of eight Gram-positive bacteria (*L. lactis*, *S. epidermidis*, *B. subtilis*, *R. opacus*, *C. xerosis*, *C. glutamicum* ATCC 13032, and *C. glutamicum* BE) and also in three out of eight Gram-negative bacteria (*P. putida*, *K. pneumoniae*, and *E. coli* DH5α). The MicC-EcHfq system allowed >50% of target gene repression achieved in six bacteria (*C. glutamicum* ATCC 13032, *S. coelicolor*, *E. coli* DH5α, *C. necator*, *V. natriegens*, and *A. hydrophila*). The generally improved knockdown effect of the BHR-sRNA system in Gram-positive bacteria might be explained by the evolutionary location of *B. subtilis* closer to many of the Gram-positive bacteria (Fig. 2a). In addition, the GC content of the RoxS scaffold (51.4%) is higher than that (43%) of the previously developed *E. coli* MicC scaffold which might have affected the scaffold stability, thus improving the knockdown effect in some Gram-negative bacteria (Supplementary Fig. 4). Further studies will be needed to understand the exact mechanisms affecting the knockdown efficiencies.

## Mitigation of virulence-associated phenotypes by target gene knockdown in pathogenic bacteria

The increased emergence of multidrug-resistant pathogenic bacteria coupled with the antibiotic discovery seriously lagging behind during the last several decades have exacerbated the already urgent antimicrobial resistance crisis[34]. As many pathogenic bacteria rely on sRNA-mediated regulations for host infection[35], we sought to hijack this machinery and transplant synthetic sRNAs targeting virulence factors for the treatment of these pathogens. The BHR-sRNA system was thus applied to a clinically isolated *S. epidermidis*, a Gram-positive skin commensal bacterium which is also an opportunistic pathogen causing hospital-acquired infections in individuals with compromised immunity[36]. Formation of biofilms, which increases resistance to antibacterial agents and to the host defense, is one of its primary virulence factors. Thus, two target genes, *icaA* which is the first gene in the *icaADBC* operon responsible for biofilm formation[37] and *sarA* encoding a positive regulator of the *ica* operon[38], were chosen as knockdown targets. Knockdown of *icaA* and *sarA* was similarly efficient, resulting in 72.8% ($P = 6.8 \times 10^{-6}$) and 73.0% ($P = 4.7 \times 10^{-7}$) reduction in biofilm formation, respectively, when compared with that by the strain without sRNA (Fig. 2c).

Next, the knockdown efficiency of the BHR-sRNA system was examined in *K. pneumoniae*, a representative Gram-negative multidrug-resistant pathogen which causes serious infectious diseases including pneumonia, sepsis, urinary tract infections, and bloodstream infections[39]. The knockdown target *folA*, an essential gene encoding dihydrofolate reductase, was selected for its role in the susceptibility of *K. pneumoniae* to the antibiotic trimethoprim[8]. The viability of *K. pneumoniae* harboring the anti-*folA* sRNA in test tube-scale culture in the presence of different concentrations of trimethoprim significantly decreased when compared with that without sRNA (Fig. 2d). In the presence of 1 µg mL$^{-1}$ of trimethoprim, *K. pneumoniae* harboring the

anti-*folA* sRNA showed 57.5% decreased cell growth (OD$_{600}$) compared with that without sRNA ($P = 2.5 \times 10^{-3}$). Thus, the synthetic sRNA technology using the BHR-sRNA system allowed mitigation of virulence-associated phenotypes in both representative Gram-positive and Gram-negative pathogenic bacteria. Based on these results, many potential applications including in vivo microbiome engineering are envisioned by integrating the synthetic sRNA technology with the appropriate in vivo RNA delivery systems (e.g., specific phage, plasmid, or gold nanoparticles)[40].

## Rapid prototyping of gene knockdown targets for the enhanced production of chemicals

In developing industrial strains by metabolic engineering, it is important to examine the effects of amplifying and repressing single and multiple genes on the performance of the strain. Unlike large-scale gene amplification tests, which can be done relatively easily by plasmid-based overexpression, the knockout or knockdown experiments on many genes are rather difficult. We previously demonstrated in *E. coli* that the sRNA-based knockdown strategy can be a solution[11,41]. Thus, we examined whether the BHR-sRNA system can be used as a widely applicable gene knockdown strategy in various bacteria.

As an example knockdown target, the type III polyketide synthase RppA from *Streptomyces griseus* capable of producing a red-colored pigment flaviolin was selected[41] (Supplementary Note 3, Supplementary Fig. 3b). Knockdown of heterologous *rppA* using the BHR-sRNA system resulted in successful reduction of flaviolin production in *E. coli* (Supplementary Fig. 3c). Knockdown of *rppA* by the BHR-sRNA system in *R. opacus* resulted in reduced flaviolin production as well as reduced cell growth (Supplementary Fig. 3d). Next, two endogenous genes, *lysA* encoding diaminopimelate decarboxylase and *pyc* encoding pyruvate carboxylase, in *C. glutamicum* were knocked down by BHR-sRNA, resulting in expected alterations in the corresponding phenotypes (Supplementary Fig. 3e–h, Supplementary Note 3). These results demonstrate that the BHR-sRNA system can be used as a plasmid-based knockdown gene target screening tool, eliminating the need for laborious and time-consuming genome engineering. The real-world applications of the BHR-sRNA system in metabolic engineering of *C. glutamicum*, one of the most important and widely used industrial strains, are showcased in the following sections.

First, the BHR-sRNA system was applied to knockdown rationally selected target genes to enhance the production of valerolactam, a cyclic form of ω-amino acid 5-aminovaleric acid, used as a monomer for the production of various polymers including polyamide-5 and polyamide-6,5. Based on the strategy developed for valerolactam production in *E. coli* from glucose[42], the *act* gene encoding β-alanine CoA transferase from *Clostridium propionicum* was introduced into the *C. glutamicum* AVA2 strain capable of producing 5-AVA[43], resulting in the VL strain. Shake flask culture of the VL strain produced 4.01 g L$^{-1}$ of valerolactam (Fig. 3a, b, Supplementary Fig. 5a).

Based on previous literature studies carried out to enhance the production of L-lysine and 5-AVA, two precursors of valerolactam, 12 genes were chosen as potential knockdown targets for the enhanced production of valerolactam (Fig. 3b and Supplementary Table 6). Shake flask cultivation of the 12 strains each harboring different target-specific sRNAs showed that three knockdown targets significantly enhanced the production of valerolactam (Fig. 3b, c): *gdh* encoding glutamate dehydrogenase (5.08 g L$^{-1}$; $P = 9.3 \times 10^{-4}$), *hom* encoding homoserine dehydrogenase (4.56 g L$^{-1}$; $P = 2.1 \times 10^{-3}$), and *icd* encoding isocitrate dehydrogenase (4.49 g L$^{-1}$; $P = 3.7 \times 10^{-3}$). Western blot analysis revealed that the levels of proteins encoded by *gdh* and *icd* were repressed by 93.3% ($P = 3.0 \times 10^{-4}$) and 40.0% ($P = 0.048$), respectively, by the target-specific sRNAs (Supplementary Fig. 5b–d; see Methods for details). The protein encoded by *hom* could not be identified by western blot analysis (see source data for Supplementary Fig. 5c, d). Taken together, these results demonstrate not only the first

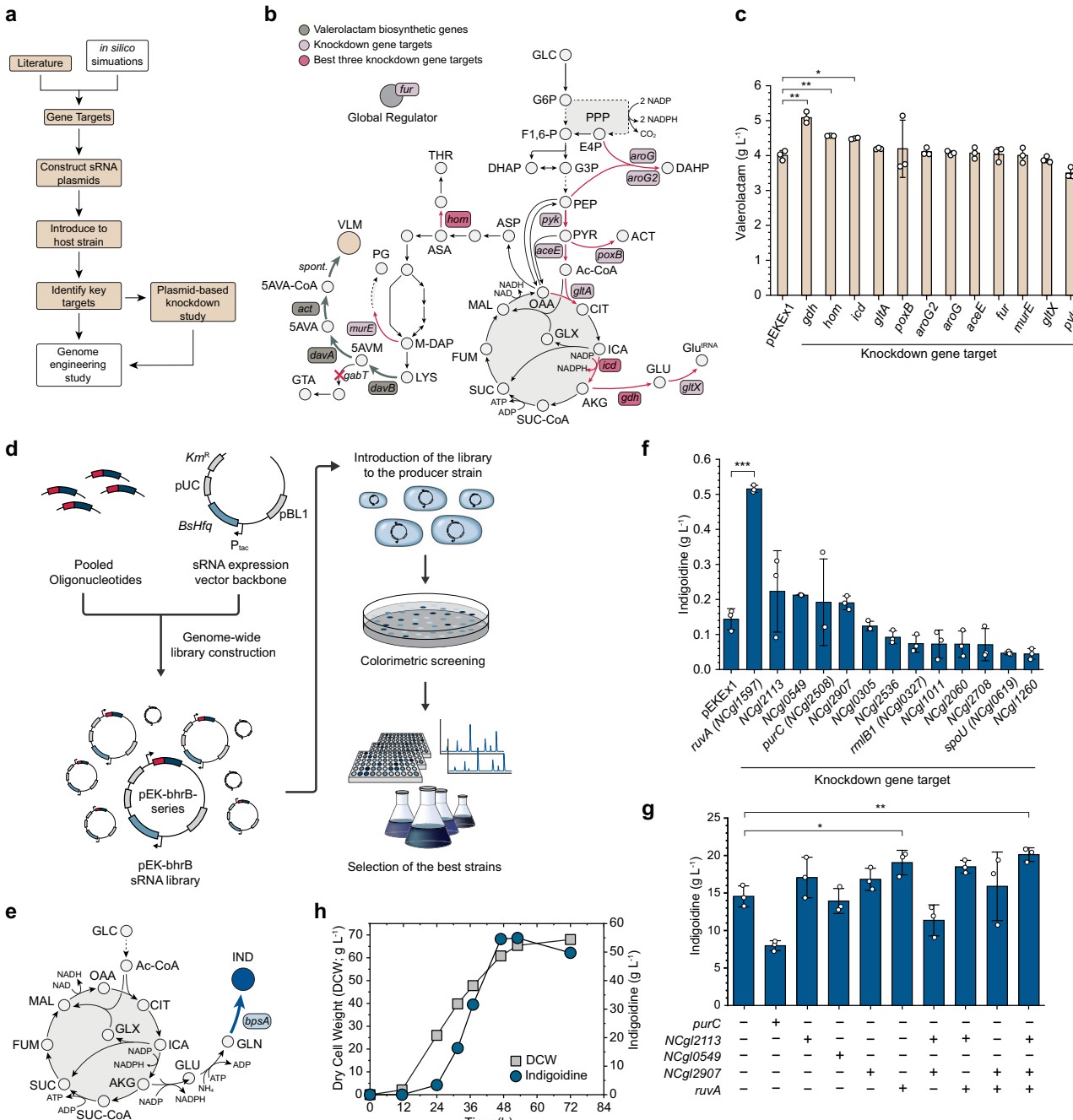

**Fig. 3 | Rapid identification of gene knockdown targets for enhanced chemical production. a** Workflow to identify key target genes to enhance valerolactam production in *C. glutamicum*. Steps employed for the enhanced production of valerolactam are described in filled rectangles. Additional steps that would facil-itate high-performing strain construction are described in empty rectangles, which are exemplified in Supplementary Note 4 for methyl anthranilate production. **b** Metabolic pathways for the biosynthesis of valerolactam, illustrating target genes selected (light pink boxes) and experimentally validated target genes that showed enhanced valerolactam production. Red 'X' denotes gene knockout; Red arrows represent reactions selected for knockdown. Dark green arrows represent het-erologous reactions introduced. Abbreviations in Supplementary Note 5. **c** Valer-olactam production titers obtained by the knockdown of the screened 12 gene targets. Error bars, mean ± SD ($n$ = 3 biologically independent samples). *$P$ < 0.017, **$P$ < 0.0033, ***$P$ < 0.00033, determined by two-tailed Student's *t*-test. *P*-value thresholds were adjusted using Bonferroni correction ($P < \alpha/m$). Enzymes that encode the genes listed are provided in Supplementary Table 6. **d** Schematic

workflow describing the colorimetric screening of strains in high-throughput mode using synthetic sRNA library. **e** Metabolic pathways for the biosynthesis of indi-goidine. **f** Indigoidine production by *C. glutamicum* WT-BpsA introduced with 13 re-cloned BHR-sRNA plasmids (harboring beneficial sRNAs screened from approximately 68,700 colonies; Supplementary Fig. 7a). Error bars, mean ± SD ($n$ = 3 biologically independent samples). ***$P$ < 0.005, determined by two-tailed Student's *t*-test. Enzymes that encode the genes listed are provided in Supple-mentary Table 8. **g** Indigoidine production by BIRU11 introduced with the combi-nation of single, double, or triple knockdown of five genes selected, where target genes were knocked down by start codon exchange in the chromosome. Error bars, mean ± SD ($n$ = 3 biologically independent samples). *$P$ < 0.025, **$P$ < 0.005, determined by two-tailed Student's *t*-test. *P*-value thresholds were adjusted using Bonferroni correction ($P < \alpha/m$). **h** Fed-batch fermentation profile of the final indigoidine producer BIRU20. Symbols: blue circle, indigoidine (g L$^{-1}$); grey square, dry cell weight (g L$^{-1}$). Source data are provided as a Source Data file.

production of a lactam in *C. glutamicum* from glucose, but also the highest valerolactam production reported.

Additionally, we combined the BHR-sRNA system together with flux balance analysis simulation (Supplementary Data 2, 3) to screen for gene targets in another *C. glutamicum* strain capable of producing methyl anthranilate (MANT), a grape flavoring compound[44] (Supplementary Fig. 6). First, we validated three target genes (*gnd*, *tkt*, and *pgl*) that improved MANT production titers in *C. glutamicum* harboring MANT biosynthetic genes in one plasmid and the BHR-sRNA system in another plasmid (Supplementary Fig. 6c; Supplementary Table 7, Supplementary Note 4). Next, we examined whether the beneficial effects of gene knockdown on MANT production can be translated into genome engineering so that sRNA plasmid-free strains can be developed. The three targets identified above were engineered by changing the start codon of the chromosomal target genes from ATG to GTG or TTG to endow gene knockdown effect (Supplementary Fig. 6d). Shake flask culture of the resultant strains demonstrated 16% increase in the MANT production titers from 192 to 223 mg L$^{-1}$ in the final engineered *C. glutamicum* strain in which *gnd* was knocked down (harboring two plasmids containing genes for MANT production; Supplementary Fig. 6e). These results suggest that the BHR-sRNA system can be employed as a powerful tool for rapidly identifying gene knockdown targets followed by developing plasmid-free microbial cell factories.

### Colorimetric screening of indigoidine producers using genome-scale synthetic sRNA library

A genome-scale sRNA library can be used to rapidly identify gene knockdown targets, including non-obvious targets, especially when combined with observable phenotypes such as fluorescence or color. To exploit this capability, a genome-wide BHR-sRNA library covering 2959 *C. glutamicum* genes was constructed: the first 24 nt sequences of all coding gene were prepared (Supplementary Data 4) and cloned into pEK-bhrB to prepare a pooled library (Fig. 3d, Methods).

Indigoidine is a non-ribosomal peptide and a promising natural colorant with a wide range of potential applications in the cosmetic, dye, and textile industry[45]. We aimed at rapidly developing an indigoidine producing *C. glutamicum* strain by colorimetric screening using synthetic sRNAs. As indigoidine exhibits deep blue color, simple screening of indigoidine overproducing strains is possible even with the naked eye, eliminating the need for complex equipment for analysis. First, indigoidine producing *C. glutamicum* WT-BpsA strain was constructed by expressing *bpsA* from *Streptomyces lavendulae* encoding phosphopantetheinylation-dependent non-ribosomal peptide synthetase (Fig. 3e). Then, the genome-scale BHR-sRNA library was transformed into WT-BpsA for high-throughput colorimetric screening. Among 68,700 colonies (corresponding to 23-fold the size of the sRNA library) obtained, 108 colonies that were significantly darker in color were selected for further characterization of the affected genes (See Methods for details). Among 108 colonies, 80 colonies showed improved indigoidine titers compared with that obtained with the control WT-BpsA strain (Supplementary Fig. 7a) and 13 colonies showed over 5-fold increase in indigoidine titer.

Among 108 initially screened colonies, there were duplicate colonies for six knockdown gene targets (*NCgl109*, *NCgl574*, *NCgl755*, *NCgl1496*, *NCgl1540*, and *NCgl2427*), and triplicate colonies for one knockdown gene target (*NCgl1893*) (Supplementary Fig. 7a). However, some of these colonies harboring identical sRNAs showed different indigoidine production levels, indicating colony variations in the initial screening stage. Therefore, to confirm that the improvement of indigoidine titer was indeed due to the knockdown of the target genes rather than colony variation, we reconstructed BHR-sRNA plasmids for the top 13 gene targets (Supplementary Table 8) apart from the pooled library and reintroduced to WT-BpsA (Supplementary Data 5) for cultivation in biotin-limited CGXII minimal medium. Five

targets, *NCgl2113*, *ruvA* (*NCgl1597*), *NCgl0549*, *purC* (*NCgl2508*), and *NCgl2907*, upon knockdown were found to have significant effects on increasing indigoidine production (Fig. 3f). The highest indigoidine titer (0.515 g L$^{-1}$) could be achieved by *ruvA* knockdown, which corresponds to 3.6-fold increase compared with that (0.144 g L$^{-1}$) obtained with the control strain ($P = 3.5 \times 10^{-5}$; Fig. 3f). Interestingly, the five knockdown targets identified are not involved in the mainstream metabolic pathway for indigoidine production, but are rather involved in membrane components (*NCgl0549*, *NCgl2907*), genetic recombination and DNA repair (*ruvA*), IMP production (*purC*), or is hypothetical (*NCgl2113*). These results emphasize the importance of using a genome-scale sRNA library in screening all genes allowing identification of non-obvious gene knockdown targets that are difficult to be rationally found.

After identifying the five best knockdown target genes by sRNA-based screening, we tested the construction of sRNA plasmid-free strains by changing the start codon of the chromosomal target genes (from ATG to GTG or TTG, as described earlier for MANT production) in an engineered indigoidine strain (BIRU-NP) we recently developed[45] (Supplementary Fig. 7b). Shake flask culture of the five resultant strains revealed that *ruvA*-knockdown resulted in the highest indigoidine titer (19.1 g L$^{-1}$; $P = 0.022$; Fig. 3g), in agreement with the results from the sRNA plasmid-harboring strains (Fig. 3f). In addition, the top three (*NCgl2113*, *ruvA*, and *NCgl0549*) of the five knockdown targets were combinatorially knocked down by start codon exchange. Knockdown of all three genes (the BIRU20 strain) further increased indigoidine titer to 20.1 g L$^{-1}$, which corresponds to 38.3% increase compared with that (14.6 g L$^{-1}$) obtained with the control BIRU11 strain ($P = 4.4 \times 10^{-3}$; Fig. 3g). Western blot analysis revealed that the protein expression level of RuvA, the knockdown target found to be primarily responsible for the enhanced indigoidine production, was reduced by 80.4% ($P = 1.87 \times 10^{-4}$; Supplementary Fig. 7c, d) by the start codon exchange in the genome. Since RuvA is a DNA helicase responsible for accelerating genomic recombination, knockdown of *ruvA* would have resulted in improved genetic stability of the indigoidine-producing strain due to the reduced homologous recombination. However, the exact mechanism on how knocking down *ruvA* resulted in improved indigoidine production requires further study. We also tested the combinatorial knockdown of the top three (*NCgl2113*, *ruvA*, and *NCgl0549*) targets by introduction of plasmids harboring two or three sRNAs (Supplementary Fig. 7e). Although simultaneous knockdown of all the three genes did not lead to the highest production, three out of four sRNA combinations resulted in higher production of indigoidine when compared with that by the *ruvA*-knockdown strain (Supplementary Fig. 7e).

Finally, fed-batch cultures were carried out to assess the performance of the engineered strain in a 6.6-L bioreactor. The pH-stat fed-batch fermentation of the triple-knockdown BIRU20 strain resulted in the production of indigoidine to a final titer of 54.9 g L$^{-1}$ (Fig. 3h, Supplementary Fig. 7f, g), an 11.4% increase from the highest indigoidine titer (49.3 g/L) reported recently[45]. The productivity, content, and yield were also all increased with the tradeoff of decreased maximum specific growth rate (Supplementary Table 9). An independently performed fed-batch fermentation produced 53.2 g L$^{-1}$ of indigoidine (Supplementary Fig. 7g, h), showing good reproducibility.

## Discussion

We developed a synthetic sRNA system, BHR-sRNA, for the convenient and efficient knockdown of genes in diverse bacteria by employing RoxS and Hfq from *B. subtilis*. The BHR-sRNA system was able to knock down reporter genes in 16 diverse bacterial species comprising pathogenic, commensal, probiotic, or industrial strains, demonstrating wide applicability of this sRNA technology. The BHR-sRNA system

was implemented to knockdown virulence factors for the treatment of multidrug-resistant pathogenic bacteria, *K. pneumoniae* and *S. epidermidis*, suggesting potential applications in microbiome engineering[2]. The BHR-sRNA system was also employed for rapidly identifying gene knockdown targets for the enhanced production of chemicals. Rapid identification of gene knockdown targets allowed relatively easier development of engineered strains capable of improved production of valerolactam (a bulk chemical) and methylanthranilate (a fine chemical) in *C. glutamicum*, with the highest level of valerolactam production through biological means achieved. In addition, a genome-scale BHR-sRNA library targeting 2959 *C. glutamicm* genes was rapidly constructed (within 2–3 days) for screening all known genes and used for identifying gene knockdown targets for the enhanced production of indigoidine as an example product. It should be noted that sRNA technology allowed identification of non-obvious gene knockdown targets beneficial for enhanced indigoidine production, which could not have been possible by rational examination of the metabolic pathways. The gene knockdown targets identified were translated into genome-engineered versions by chromosomal manipulation of the start codon to develop plasmid-free industrially more friendly strains. The indigoidine producer developed this way was able to produce indigoidine to the highest titer reported to date.

As the BHR-sRNA system works well in a wide range of bacteria, a similar strategy can be applied in other bacteria for developing high performance strains, with slight modifications specific for each bacterium. For those bacteria showing relatively lower knockdown efficiencies (Fig. 2b), further strain-dependent optimization of the sRNA platform will enable more efficient knockdown. Such strategies include altering the promoter strength[46], codon optimization of the *BsHfq* gene, and employing plasmids with different copy numbers[47] that have been previously demonstrated in *E. coli*. In addition, the BHR-sRNA can be employed together with CRISPRi for the dual transcriptional and translational repression of target genes, which was shown to result in more efficient knockdown, albeit at the cost of reduced cell growth (Supplementary Fig. 2). Another potential strategy is engineering the Hfq protein, as Hfq is known to aid the binding of sRNA to the target mRNA[48]. Although the applicability of the BHR-sRNA system was demonstrated in 16 different bacteria in this study, it is anticipated that the system will also be readily adaptable to other non-model bacteria without precedent genetic engineering tools. Taken together, this study demonstrates that the BHR-sRNA platform is a generally applicable synthetic biology and metabolic engineering toolkit for rapid, efficient, accurate, and high-throughput knockdown of gene targets in diverse bacteria.

## Methods

### Materials and strains
Valerolactam and MANT were purchased from Sigma-Aldrich. Anti-6X His tag antibody (HRP) was from Abcam, where it was diluted 5000-fold for western blot analysis. Indigoidine was obtained from the previous study[45]. All bacterial strains and plasmids used in this study are listed in Supplementary Data 5. *E. coli* DH5α (Invitrogen) was used for general cloning work.

### Routine media
Lysogeny broth (LB) (10 g L⁻¹ NaCl, 10 g L⁻¹ tryptone, and 5 g L⁻¹ yeast extract) and LB agar [LB supplemented with 1.5% (w/v) agar] were used for cloning works. When required, 25 μg mL⁻¹ of kanamycin (Km) or 200 μg mL⁻¹ spectinomycin (Spc) (100 μg mL⁻¹ for *E. coli* strains) were added for selection procedures. For routine work with *C. glutamicum* strains, BHIS media containing 91.1 g L⁻¹ sorbitol and 37 g L⁻¹ brain heart infusion (BHI) was used. NCM media was used for the preparation of *C. glutamicum* electrocompetent cells[49]. For transformation of *B. subtilis*, HS [0.2 g L⁻¹ (NH₄)₂SO₄, 1.4 g L⁻¹ K₂HPO₄, 0.6 g L⁻¹ KH₂PO₄, 0.1 g L⁻¹ sodium citrate dihydrate, 0.02 g L⁻¹ MgSO₄·7H₂O, 5 g L⁻¹ glucose,

0.2 g L⁻¹ casamino acid, 5 g L⁻¹ yeast extract], LS [0.2 g L⁻¹ (NH₄)₂SO₄, 1.4 g L⁻¹ K₂HPO₄, 0.6 g L⁻¹ KH₂PO₄, 0.1 g L⁻¹ sodium citrate dihydrate, 0.02 g L⁻¹ MgSO₄·7H₂O, 5 g L⁻¹ glucose, 0.1 g L⁻¹ casamino acid, 1 g L⁻¹ yeast extract, 2.5 mM MgCl₂, 0.5 mM CaCl₂], LB agar, and MR + Trp agar [6.67 g L⁻¹ KH₂PO₄, 4 g L⁻¹ (NH₄)₂HPO₄, 0.8 g L⁻¹ citric acid, 50 mg L⁻¹ tryptophan, 5 mL l⁻¹ trace metal solution, 1.5% (w/v) agar] were used. In trace metal solution, following components were included per liter: 2 g CaCl₂, 2.2 g ZnSO₄·7H₂O, 0.58 g MnSO₄·5H₂O, 1 g CuSO₄·5H₂O, 0.1 g (NH₄)₆Mo₇O₂₄·4H₂O, 0.02 g Na₂B₄O₇·10H₂O, 10 g FeSO₄·7H₂O, 5 mL fuming HCl aqueous solution. When required, 5 μg mL⁻¹ of chloramphenicol (Cm) was added. For transformation of *S. coelicolor*, mannitol soya flour (MS) medium agar (20 g mannitol, 20 g soya flour, and 20 g agar per 1 L of tap water) supplemented with 10 mM of MgCl₂ was used. When required, 50 μg mL⁻¹ of apramycin, 30 μg mL⁻¹ of nalidixic acid, and 5 μg mL⁻¹ of thiostrepton were added.

### Construction of plasmids
Standard protocols were used for PCR, gel electrophoresis and transformation experiments[50]. The oligonucleotides used in this study are listed in Supplementary Data 6. Polymerases used for PCR reactions were either Lamp-Pfu or Pfu purchased from Biofact. Restriction endonucleases and T4 polynucleotide kinase (PNK) were purchased from either Enzynomics or NEB. Detailed methods for the construction of each plasmid used in this study are described in Supplementary Method 1.

### Flow cytometry
Samples for flow cytometry were prepared and analyzed as described previously in ref. 49. Briefly, the *C. glutamicum* strains for sRNA knockdown were inoculated to 50 mL falcon tubes containing 5 mL BHIS media containing Km (25 μg mL⁻¹) and Spc (200 μg mL⁻¹) and cultivated at 30 °C for 25 h with agitation at 200 rpm. Cells were harvested by centrifugation at 13,200 rpm, washed with phosphate-buffered saline (PBS) and resuspended with the same buffer for flow cytometry analysis using fluorescence-activated cell sorting (FACS; MoFlo XDP, Beckman Coulter, Inc., Miami, FL, USA) based on high fluorescence intensity detection through a 530/40 band-pass filter for the GFP emission spectrum (Fig. 1d, Supplementary Fig. 1b–d). BD FACS LSRFortessa cell analyzer (BD Biosciences, NJ, USA) installed at the KAIST Bio Core Center was also used through a 530/30 band-pass filter for the GFP emission spectrum (Fig. 1d, Supplementary Fig. 2a; Supplementary Fig. 8).

### RT-qPCR analysis
The *C. glutamicum* strains for sRNA knockdown were inoculated to 50 mL falcon tubes containing 5 mL BHIS media containing Km (25 μg mL⁻¹) and Spc (200 μg mL⁻¹) and cultivated at 30 °C for 25 h with agitation at 200 rpm. Total RNA extraction was performed using Fast HQ RNA Extraction Kit (Invitrogen) and following the manufacturer's protocol. The extracted total RNA samples were used as templates for reverse transcription qPCR (RT-qPCR). The *dnaK* gene was selected as the housekeeping gene (control) which was amplified using primers dnaK_RT_F and dnaK_RT_R. The *GFP* gene was amplified using primers GFP_RT_F and GFP_RT_R. RT-qPCR was performed using PrimeScript RT-PCR Kit (Takara Bio) and LightCycler® 96 real-time PCR system (Roche) according to the manufacturer's protocols. The relative abundance of mRNAs of reporter genes was quantified on the basis of the cycle threshold (Ct) value and was calculated using the $2^{-\Delta\Delta Ct}$ method.

### Knockdown of reporter genes in diverse bacteria
The construction of plasmids, transformation methods (electroporation/conjugation), and cultivation conditions used in the knockdown study for each bacterium are described in Supplementary Method 2, and the types of reporter and the plasmids used for each bacterium are

summarized in Supplementary Table 5. All optical density (OD) and fluorescent reporter measurements were carried out using Spark Multimode Microplate Reader (Tecan, Männedorf, Switzerland) with fluorescent intensities (excitation 485 nm, emission 535 nm, gain 100 for GFP and EGFP; excitation 450 nm, emission 600 nm, gain 100 for RFP; excitation 560 nm, emission 610 nm, gain 147 for mCherry) in black, clear-bottom 96-well plates (SPL, Korea). Confirmation of target gene knockdown was also obtained through western blot analysis.

### *Klebsiella pneumoniae* trimethoprim susceptibility assay

*K. pneumoniae* strains harboring pEKEx1 or pEK-bhr-anti-folA were inoculated to test tubes containing 5 mL LB medium supplemented with 250 µg mL$^{-1}$ Km and cultured overnight at 30 °C with rotary shaking at 200 rpm. The strains were then inoculated to test tubes containing 2 mL of fresh LB medium at varying concentrations of trimethoprim (0, 0.05, 0.1, 0.5, and 1 µg mL$^{-1}$) and cultured at 30 °C with rotary shaking at 200 rpm for 24 h. At the end of the cultivation, OD$_{600}$ was measured to determine cell viability.

### *Staphylococcus epidermidis* biofilm assay

To test the effect of *icaA* or *sarA* knockdown on *S. epidermidis* biofilm formation, biofilm quantification was performed[51]. Briefly, the *S. epidermidis* strains harboring pRMC2, pRMCbhr-anti-icaA, or pRMCbhr-anti-sarA were inoculated to 5 mL TSB media in 50 mL falcon tubes supplemented with 5 µg mL$^{-1}$ of Cm. After overnight culture at 37 °C, 200 rpm, the cells were transferred to each well (initial OD$_{600}$ adjusted to 0.2) in a 96 transparent well plate (SPL) containing 100 µL of TSB supplemented with 5 µg mL$^{-1}$ of Cm, 1 µg mL$^{-1}$ of anhydrotetracycline (ATC) for sRNA expression, and 0.4% glucose for biofilm induction. The well plate containing the inocula was incubated in a static 37 °C incubator for 48 h. The growth (OD$_{600}$) of the planktonic bacteria was measured after shaking the well plate and transferring the cultures to other wells. The wells containing the biofilms were washed twice with 150 µL of PBS. After aspiration of the remaining PBS, the remaining biofilms were stained with 150 µL of 0.1% (v/v) crystal violet solution for 10 min at room temperature. Then, the solution was discarded and the wells were washed with PBS three times. To solubilize the crystal violet and thus to quantify the biofilms, 150 µL of 33% (v/v) acetic acid was added to each well and was incubated for 10 min at room temperature. After shaking the plate, biofilms were quantified by measuring the OD$_{570}$ values using Spark Multimode Microplate Reader (Tecan).

### *C. glutamicum* genome manipulation

Knockdown of genes *gnd*, *pgl*, and *tkt* in *C. glutamicum* DBDH strain (a MANT-producing strain) and of genes *NCgl2113*, *ruvA*, *NCgl0549*, *purC*, and *NCgl2907* in *C. glutamicum* BIRU11-NP were performed with a marker-free system using *Bacillus subtilis sacB* gene via two rounds of recombination[44]. Transformations of *C. glutamicum-E. coli* shuttle vectors were performed by electroporation followed by heat shock[49]. For detailed information of the construction of each engineered *C. glutamicum* strain used in the study, refer to Supplementary Method 3.

### Media and culture conditions

For the RppA knockdown assay in *R. opacus*, the strains were inoculated to 5 mL LB media containing 10 g L$^{-1}$ of glucose contained in 50 mL conical tubes supplemented with appropriate antibiotics and incubated at 30 °C with rotary shaking at 200 rpm until the OD$_{600}$ of the cells reached ~4. Then, 200 µL of the culture was transferred to fresh 10 mL LB media containing 10 g L$^{-1}$ of glucose contained in 50 mL conical tubes supplemented with appropriate antibiotics and incubated at 30 °C until the OD$_{600}$ of the cells reached ~1. Then, 1 mM of isopropyl β-D-1-thiogalactopyranoside (IPTG) and 0.17 M of acetamide were added to induce expression of *BsHfq* and *rppA*, respectively. The cells were additionally grown for 24 h, after which analysis of flaviolin was performed.

For the RppA knockdown assay in *E. coli* DH5α strains, the strains were inoculated to 5 mL LB media contained in 50 mL conical tubes supplemented with appropriate antibiotics and incubated at 37 °C with rotary shaking at 200 rpm overnight. Then, 200 µL of the culture was transferred to fresh 10 mL LB media contained in 50 mL conical tubes supplemented with appropriate antibiotics and 1 mM of IPTG (for *BsHfq* expression) and incubated at 30 °C for 36 h, after which analysis of flaviolin was performed. When measuring the OD$_{600}$ values of the *R. opacus* and *E. coli* strains harboring *rppA*, the cultures were centrifuged and the pellets were resuspended in equal volumes of PBS to eliminate the interference of flaviolin on the OD$_{600}$ measurement for cell growth.

For valerolactam production, *C. glutamicum* cultures were performed in baffled flasks (300 mL) containing 25 mL GLP media. GLP media (pH 7.0) comprises per liter: 80 g glucose, 1 g K$_2$HPO$_4$, 1 g KH$_2$PO$_4$, 1 g urea, 20 g (NH$_4$)$_2$SO$_4$, 10 g yeast extract, 60 g CaCO$_3$, 1 g MgSO$_4$, 50 mg CaCl$_2$, 100 µg biotin, 10 mg β-alanine, 10 mg thiamine·HCl, 10 mg nicotinic acid, 1.3 mg (NH$_4$)$_6$MoO$_{24}$, 10 mg FeSO$_4$, 10 mg MnSO$_4$, 5 mg CuSO$_4$, 10 mg ZnSO$_4$, and 5 mg NiCl$_2$. Cells were inoculated from glycerol stocks to test tubes containing 5 mL of BHIS medium and cultivated in shaking incubator at 200 rpm and 30 °C for 16 h before being transferred to the flask culture containing GLP media, which was cultivated at 200 rpm and 30 °C for 48 h.

*C. glutamicum* shake flask cultures for MANT production were performed in baffled flasks (300 mL) containing 25 mL of CGXII minimal medium supplemented with 40 mg L$^{-1}$ of L-tryptophan. The CGXII minimal medium comprises per liter: 40 g glucose, 20 g (NH$_4$)$_2$SO$_4$, 5 g urea, 1 g KH$_2$PO$_4$, 1 g K$_2$HPO$_4$, 0.25 g MgSO$_4$·7H$_2$O, 42 g 3-morpholinopropanesulfonic acid (MOPS), 13 mg CaCl$_2$·2H$_2$O, 10 mg FeSO$_4$·7H$_2$O, 14 mg MnSO$_4$·5H$_2$O, 1 mg ZnSO$_4$·7H$_2$O, 0.3 mg CuSO$_4$·5H$_2$O, 0.02 mg NiCl$_2$·6H$_2$O, 30 mg protocatechuic acid, 0.5 mg thiamine and 0.5 mg biotin. To prepare seed culture, glycerol cell stock was inoculated into a test tube containing 5 mL of BHIS medium and cultivated in a shaking incubator at 200 rpm and 30 °C for 14 h. Then, 1 mL of the seed culture was used to inoculate the flask culture, which was cultivated at 200 rpm and 30 °C. After 6 h of cultivation, 5 mL of tributyrin and 1 mM of IPTG were added. The flask culture was terminated at 48 h for analysis.

For indigoidine production using wild-type *C. glutamicum*, cultures were done in test tubes (15 mL) containing 2 mL of biotin-limited CGXII minimal medium where biotin supplementation was reduced to 0.004 mg L$^{-1}$. Cells were inoculated to test tubes containing 2 mL of BHIS medium and cultivated in a shaking incubator at 200 rpm and 30 °C for 14 h. Then, 50 µL of the seed culture was inoculated to test tubes containing 2 mL of biotin-limited CGXII minimal medium, and cultivated in a shaking incubator at 200 rpm and 30 °C for 48 h (1 mM IPTG induction at 6 h). For indigoidine production using engineered *C. glutamicum* BIRU11 and subsequent engineered derivatives, cultures were done in 250 mL baffled flasks containing 25 mL of GAP medium comprising of 70 g L$^{-1}$ glucose, 40 g L$^{-1}$ (NH$_4$)$_2$SO$_4$, 1 g L$^{-1}$ KH$_2$PO$_4$, 0.4 g L$^{-1}$ MgSO$_4$·7H$_2$O, 0.01 g L$^{-1}$ FeSO$_4$, 0.01 g L$^{-1}$ MnSO$_4$, 4 µg L$^{-1}$ biotin, 200 µg L$^{-1}$ thiamine·HCl, and 15 mL L$^{-1}$ commercial soy sauce (Soy Sauce Jin Gold F3, Sempio, Gyeonggi, Korea).

### Fed-batch fermentations for indigoidine production

Fed-batch fermentations were performed in a 6.6 L jar fermenter (Bioflo 320; New Brunswick Scientific Co., Edison, NJ) for indigoidine production, containing 1.8 L of GAP medium. Preseed culture was prepared by inoculating *C. glutamicum* cells from BHIS plate into a 250 mL shake flask containing 20 mL of BHIS medium supplemented with 20 g L$^{-1}$ glucose, cultivated in a rotating shaker at 200 rpm and 30 °C for 12 h. The preseed was transferred (5% inoculation v/v) into four 250 mL shake flasks each containing 25 mL of seed medium comprising 40 g L$^{-1}$ glucose, 20 g L$^{-1}$ (NH$_4$)$_2$SO$_4$, 1 g L$^{-1}$ KH$_2$PO$_4$, 0.4 g L$^{-1}$ MgSO$_4$·7H$_2$O, 0.01 g L$^{-1}$ FeSO$_4$, 0.01 g L$^{-1}$ MnSO$_4$, 50 µg L$^{-1}$ biotin,

200 µg L$^{-1}$ thiamine·HCl, 10 g L$^{-1}$ yeast extract, and 15 mL L$^{-1}$ commercial soy sauce (Soy Sauce Jin Gold F3, Sempio, Korea). After 18 h, the seed was transferred into the fermenter. The fermentation pH was maintained at 7.0 with the addition of 28% (v/v) NH$_4$OH, and fermentation temperature was set at 30 °C. The starting agitation speed was fixed at 600 rpm, and air flow rate was kept at 2 L min$^{-1}$. The dissolved oxygen concentration (DO) was maintained at 40% of air saturation by automatically increasing the agitation speed up to 1000 rpm, and changing the percentage of pure oxygen added. Cells were induced with 1 mM IPTG when initial DO levels fell below 70%. Feed solution composed of 700 g L$^{-1}$ glucose, 80 g L$^{-1}$ (NH$_4$)$_2$SO$_4$, and 2 g L$^{-1}$ MgSO$_4$·7H$_2$O was programmed to be fed when pH levels rise above 7.04 (Supplementary Fig. 7f). For the first feed, the cells were starved ~1 h as per the previous indigoidine fed-batch fermentation study[45].

### Construction of sRNA library

The pooled oligonucleotides containing antisense sequences of the first 24 nt of all 2959 genes in *C. glutamicum* were synthesized from Twist Biosciences (San Francisco, USA). The pooled oligonucleotide was designed as follows:

5′-taatacgactcactataggg <u>GGTCTCT</u> **GTGG** NNNNNNNNNNNNNNNN NNNNNNNNN **ACAT** A<u>GAGACC</u> ggtcttgaggggtttttg-3′

Underline denotes BsaI sites, bold indicates overhang expected to be produced by BsaI digestion, and "N"s indicates target binding sequence which is reverse complementary to the first 24 nt of the target gene. To introduce the pooled oligonucleotides containing the 24 nt antisense target binding sequences, pEK-bhrB was first constructed. The pEK-bhrB plasmid contains two proximate BsaI sequences (reverse complement with each other) between the tac promoter and the RoxS scaffold, allowing easy introduction of pooled oligonucleotides by Golden Gate cloning. To construct pEK-bhrB, the sRNA construct containing BsaI sequences was amplified from pWAS-RoxS using primers M9A1_RoxS_BsaI_F and M5A_R, which was again PCR amplified using primers M9K_F and M5E_R. The amplified gene fragments were inserted to pEK-BsHfq at StuI site using Gibson assembly, followed by site-directed mutagenesis using primers pEK_BsaI_mut_F and pEK_BsaI_mut_R to eliminate the originally existing BsaI site[52]. Then, the pooled oligonucleotides were amplified using primers Oligo_F and Oligo_R, and were digested with BsaI, followed by insertion into pEK-bhrB at BsaI sites by ligation. *E. coli* DH5α was transformed with the pooled plasmid and spread onto 150 mm plates (SPL Life Sciences) containing LB agar supplemented with 25 µg mL$^{-1}$ of Km. To make sure all 2959 targets are included, number of colonies of at least 10-fold the size of the library would be necessary[41]. Therefore 36,782 DH5α colonies were collected using a cell scraper, and the plasmids were extracted using AccuPrep Nano-Plus Plasmid Maxi Extraction kit (Bioneer, Daejeon, South Korea).

### Genome-wide screening for indigoidine

The genome-scale BHR-sRNA library was transformed into WT-BpsA for high-throughput colorimetric screening. To sufficiently cover all 2959 target genes, we reasoned that a colony library size of at least 10-fold the number of target genes would be necessary as calculated based on the Monte Carlo method. Therefore, approximately 68,700 *C. glutamicum* colonies (corresponding to 23-fold the size of the sRNA library) were screened on BHIS agar plates supplemented with IPTG for *hfq* induction. Among the 68,700 colonies, we selected 108 colonies that were significantly darker in color to characterize the gene targets and to culture them for indigoidine production. The sRNA target sequence in each of the 108 colonies were identified by first amplifying the target binding region by PCR, followed by sequencing of the resulting PCR product. The initially screened indigoidine producers containing sRNA library components were inoculated to 14 mL disposable Falcon round-bottom tubes (Corning, New York, USA) containing 2 mL BHIS medium supplemented with appropriate

antibiotics. After the cells were grown at 30 °C and 200 rpm for 16 h, they were transferred to 50 mL tubes containing 5 mL biotin-limited CGXII medium supplemented with appropriate antibiotics and 1 mM IPTG. The cells were grown at 30 °C and 200 rpm for 48 h. The culture broth was diluted 10 times with DMSO, and vortexed at room temperature for 5 min for indigoidine extraction. The mixture was centrifuged and the resulting supernatant was inoculated to 96-well plates for analysis using a microarray reader (Tecan Spark, Tecan, Switzerland) with optical density measurement at 610 nm to determine relative indigoidine production.

### Analytical methods

Cell growth was measured with Ultrospec 3100 spectrophotometer (Amersham Biosciences, Uppsala, Sweden) with absorbance set at 600 nm (OD$_{600nm}$). Dry cell weight (DCW) was calculated as follows. Briefly, cell culture broth was washed with 1 M HCl solution (1:1) twice. Once the supernatant was removed, the cell pellets were dried in an oven at 70 °C for 24 h, cooled to room temperature, and subjected to a gravimetric biomass quantification using an analytical balance (CPA224S, Sartorius, Göttingen, Germany).

Glucose and organic acids were analyzed by high-performance liquid chromatography (HPLC) (Waters 1515/2414/2707, Waters, Milford, MA) equipped with a refractive index detector (2414, Waters). A MetaCarb 87H column (Agilent) was used and the mobile phase (0.01 N H$_2$SO$_4$) was flown at 0.5 mL min$^{-1}$. The column was operated at 25 °C.

For the analysis of L-lysine, 5-aminovaleric acid, and valerolactam, culture supernatant was filtered through 0.2 PVDF syringe filters (Sugentech). For detection of L-lysine and 5-aminovaleric acid, the supernatant of culture samples was reacted with *o*-phthaldehyde[42], prior to the injection into the Eclipse Zorbax-AAA column (Agilent Technologies). Linear gradient of the mobile phase A (10.0 mM Na$_2$HPO$_4$, 10.0 mM Na$_2$B$_4$O$_7$·10H$_2$O, 8.0 g L$^{-1}$ NaN$_3$, pH 8.2) and the mobile phase B (methanol, acetonitrile and water in 45:45:10 by volume) was used to separate the amino acids in the column. Borate buffer (pH 10.2, 0.40 M) was used as buffering agent instead of pH 9.0 previously described[42]. The derivatized compounds were detected using a diode array detector (DAD) at 338 nm. The column temperature was set at 25.0 °C and the flow rate of the pump was set at 0.640 mL/min. The following gradient was applied for resolving the compounds: 0–0.5 min, 0% B; 0.5–18 min, a linear gradient of B from 0% to 57%; 18–26 min, a linear gradient of B from 57 to 100%; 26–31.8 min, 100% B; 31.8–31.9 min, a linear gradient of B from 100% to 0 %; 31.9–32 min, 100% A by volume. For detection of valerolactam, mobile phase A (0.1% formic acid) and the mobile phase B (methanol) was used using the same method mentioned above. For the analysis of violacein produced from *C. violaceum* strains, the mobile phase was run at 30 °C at a flow rate of 1 mL min$^{-1}$; the mobile phase consists of solvent A (0.1% formic acid) and solvent B (acetonitrile). The following gradient was applied: 0–3 min, 10% B; 3–10 min, a linear gradient of B from 10% to 100%; 10–15 min, 100% B (all in vol%). Samples containing violacein/deoxyviolacein were monitored at 570 nm.

For the analysis of flaviolin, culture supernatant was filtered through 0.2 µm PVDF syringe filters (Sugentech). For the analysis of MANT, the separated organic phase was filtered through 0.2 µm PTFE syringe filters (Sugentech). The prepared samples were analyzed with HPLC (1260 Infinity II; Agilent Technologies, Palo Alto, CA) equipped with DAD detectors (G7115A; Agilent) and Eclipse XDB-C18 column (4.6 × 150 mm; Agilent). For the analysis of flaviolin, the mobile phase was run at 25 °C at a flow rate of 0.8 mL min$^{-1}$; the mobile phase consists of solvent A (0.1% formic acid in ddH$_2$O) and solvent B (methanol). The following gradient was applied: 0–1 min, 30% B; 1–5 min, a linear gradient of B from 30% to 70%; 5–15 min; a linear gradient of B from 70% to 95%; 15–16 min, a linear gradient of B from 95% to 100%; 16–19 min, 100% B; 19–20 min, a linear gradient of B from 100% to 30%. For analysis of MANT, the mobile phase was run at 30 °C at a flow rate

of 1.0 mL min$^{-1}$; the mobile phase consists of solvent A (0.1% tri-fluoroacetic acid in ddH$_2$O) and solvent B (acetonitrile). The following gradient was employed: 0–1 min, 10% B; 1–10 min, a linear gradient of B from 10% to 70%; 10–12 min, 70% B; 12–14 min, a linear gradient of B from 70% to 10%; 14–18 min, 10% B. Samples containing flaviolin were monitored at 254 nm; samples containing MANT were monitored at 330 nm.

Indigoidine extraction and quantification was performed in the following manner[45]. Briefly, 100 µL of the fermentation broth was mixed with 900 mL of DMSO and 2% Tween 20 solution. One hundred microliters of silica beads (0.1 and 0.5 mm, Biospec, Bartlesville, UK) was added to the mixture and vortexed for 30 s using a beadbeater (Precellys 24; Bertin Tech, Montigny-le-Bretonneux, France) at 6000 rpm. After centrifugation at 13,000 g for 1 min, 100 µL of the supernatant was collected and filtered for analysis using Spark Multimode Microplate Reader (Tecan, Männedorf, Switzerland) at OD$_{612}$. The measurements read were compared against the standard curve made previously[45] by using high purity (≥95%) indigoidine (Hangzhou Viablife Biotech, Hangzhou, China) diluted in DMSO and 2% Tween 20 solution to obtain precise concentrations.

## Statistical analysis
We did not predetermine sample sizes. All colonies were randomly selected from plates containing ~100–200 colonies and subject to independent flask culture and chemical analysis. All numerical data are presented as mean ± standard deviation (SD) from experiments done in triplicates. Means were compared using a two-tailed Student's $t$-test. $P$ values were represented $*P < 0.05$, $**P < 0.01$ or $***P < 0.001$, which were considered as significant. When multiple hypotheses were tested, the significance level thresholds were divided by the number of hypotheses, according to Bonferroni correction (corrected significance levels represented as $\alpha/m$; $\alpha$, original significance level; $m$, number of hypotheses). Thus, $*P < 0.05/m$, $**P < 0.01/m$ or $***P < 0.001/m$. The investigators were blinded to the group allocation by randomly selecting single colonies multiple times.

## Reporting summary
Further information on research design is available in the Nature Portfolio Reporting Summary linked to this article.

# Data availability
A reporting summary for this article is available as a Supplementary Information file. Data supporting the findings of this work are available within the paper and its Supplementary Information files. Source data are provided with this paper and are also available from Figshare [https://doi.org/10.6084/m9.figshare.22588612][53]. Source data are provided with this paper.

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

## Acknowledgements

The authors would like to thank Changdai Gu for MOMA and FSEOF analyses; Dr. In Jin Cho for advice on *P. putida* and *R. opacus* works; Prof. Byung-Kwan Cho and Sunkyu Hwang from Dept. Biology at KAIST for advice on *S. coelicolor* works; Prof. Ki Jun Jeong from Dept. Chemical and Biomolecular Engineering at KAIST for the plasmid pCES-I16-GFP; Jae Eun Lee and Jin Won Lee for technical assistance; Sam Mi Yu at KAIST Bio Core Center for FACS analysis. pRMC2 was a gift from Tim Foster (Addgene plasmid # 68940) and pSK9065 was a gift from Neville Firth (Addgene plasmid # 84345). This work was supported by the Development of next-generation biorefinery platform technologies for leading bio-based chemicals industry project (2022M3J5A1056072) and by Development of platform technologies of microbial cell factories for the next-generation biorefineries project (2022M3J5A1056117) from NRF supported by the Korean MSIT.

## Author contributions

S.Y.L. conceived the project. J.S.C., D.Y., and S.Y.L. designed research. J.S.C., D.Y., C.P.S.P., M.R.G., T.H., K.R.C., C.W.M., and H.Z. performed the experiments. M.R.G. contributed materials/analysis tools. J.Y.R., H.U.K., J.S.C., and D.Y. analyzed the data. S.Y.L., J.S.C., and D.Y. wrote the paper. All authors read and approved the final manuscript.

## Competing interests

S.Y.L., J.S.C., and D.Y. declare that the sRNA technology developed here is of commercial interest and have filed patents. Patent application numbers: KR 10-2019-0026219, PCT/KR2019/002715, and US 16/960,064. Other authors claim no competing interests.
