## [Peer Review File · Nature Communications]

Targeted and high-throughput gene knockdown in diverse bacteria using synthetic sRNAsEditorial Note: This manuscript has been previously reviewed at another journal that is not operating a transparent peer review scheme. This document only contains reviewer comments and rebuttal letters for versions considered at Nature Communications.

Reviewers' Comments:

Reviewer #1:

Remarks to the Author:

The authors have responded to and addressed all of this reviewer's questions.

Reviewer #2:

Remarks to the Author:

In this work, the authors first developed a synthetic sRNA scaffold platform comprising the sRNA scaffold and Hfq chaperone from *Bacillus subtilis* in *Corynebacterium glutamicum*. The authors then demonstrated the use of this platform in 16 bacterial strains and compared its efficiency to the sRNA platform developed from *Escherichia coli*. Lastly, the authors showed how the platform could be applied for virulence gene knock-down and genome-scale candidate identification in metabolic engineering for enhanced chemical production.

In general, the manuscript is well-written and easy to understand. This study addresses a technological gap for the quick experimental validation of target genes, which metabolic engineering needs. Below are my comments which require clarification before publication.

Major Comments

1. In lines 96-97, it was mentioned that all corresponding Hfq proteins were introduced with their corresponding sRNA. However, in Figure 1d, *Bacillus thuringiensis* sRNA Bts was introduced with *B. subtilis* Hfq protein instead. Please revise Figure 1d and provide data reflective of the *Bacillus thuringiensis* sRNA system.

Also, the sRNA for *Streptomyces coelicolor* is mislabeled. It should be Scr5239; Sco5239 is a gene encoding for a signal transduction histidine kinase in *S. coelicolor*. Please revise figure 1d accordingly.

2. In line 174, the authors mentioned that the BHR-sRNA system outperformed the *E. coli* system in three of eight Gram-negative bacteria. However, in Figure 2b, this is clearly obvious for only two Gram-negative strains (*Klebsiella pneumoniae* and *Pseudomonas putida*). Which is the third strain? It would be helpful to support the comparison with statistical analysis to highlight these strains in figure 2b.

3. Was the Echfq chaperone protein tested in the Gram-positive strains codon optimised? Reduced translational efficiency of the MicC-Echfq system in Gram-positive strains due to codon usage differences could be a confounding factor when comparing the sRNA systems. Please clarify and mention this in the text.

4. From Figure 2b, broad-host range performance is exhibited by both *E. coli* and *B. subtilis* sRNA-Hfq systems. In some Gram-positive strains (like the *S. coelicolor*) the *E. coli* system is more efficient than

the *B. subtilis* system (BHR-sRNA). Similar to the *E. coli* system (in supplementary table 6), the *B. subtilis* system has also been noted by the authors as more specific to Gram-positives (lines 176-177). The authors should be more specific in naming the BHR-sRNA system or qualify further why the *B. subtilis* system should be named BHR-sRNA and not *E. coli*.

5. In lines 226-227 and supplementary figure 3d, the BHR-sRNA system reduced both cell growth and flavin production in *Rhodococcus opacus*. Reduced cell growth due to the introduction of BHR-sRNA could be a confounding factor for reduced flavin. Please provide evidence of gene knockdown by BHR-sRNA of *rppA* in *R. opacus*.

Minor Comments

1. There is some incorrect labelling in Figure 2b for revision.

a. The x-axis for *Lactococcus lactis* should be labelled RoxS instead of RoxS-BsHfq as BsHfq was not expressed.

b. The x-axis for *Staphylococcus epidermidis* should be RFP and not EGFP.

Reviewer #3:

Remarks to the Author:

The authors have addressed the points I raised in my initial review. The side-by-side experimental comparison of BHR-sRNA and MicC-EcHfq strengthens the manuscript and will guide future users on which system to use depending on the host. The new results on the combination of CRISPRi and sRNA are a great addition to the manuscript and provide the community with a new strategy to reach higher silencing levels. I believe this system will be used by the community and I recommend the publication of this work.

Response to Reviewers' Comments for NMICROBIOL-22010060A

Those sentences that are changed or added are shown in red in the revised manuscript and in the SI for your convenience.

REVIEWER COMMENTS

Reviewer #1 (Remarks to the Author):

The authors have responded to and addressed all of this reviewer's questions.

[Response] Thank you very much.

Reviewer #2 (Remarks to the Author):

In this work, the authors first developed a synthetic sRNA scaffold platform comprising the sRNA scaffold and Hfq chaperone from *Bacillus subtilis* in *Corynebacterium glutamicum*. The authors then demonstrated the use of this platform in 16 bacterial strains and compared its efficiency to the sRNA platform developed from *Escherichia coli*. Lastly, the authors showed how the platform could be applied for virulence gene knock-down and genome-scale candidate identification in metabolic engineering for enhanced chemical production.

In general, the manuscript is well-written and easy to understand. This study addresses a technological gap for the quick experimental validation of target genes, which metabolic engineering needs. Below are my comments which require clarification before publication.

Major Comments

1. In lines 96-97, it was mentioned that all corresponding Hfq proteins were introduced with their corresponding sRNA. However, in Figure 1d, *Bacillus thuringiensis* sRNA Bts was introduced with *B. subtilis* Hfq protein instead. Please revise Figure 1d and provide data reflective of the *Bacillus thuringiensis* sRNA system.

[Response] Sorry for the mistake and thank you for finding this error. This was an incorrect labelling. *Bacillus thuringiensis* sRNA BtsR1 was introduced with *B.thuringiensis* Hfq (Supplementary Table 2). We have revised Figure 1d accordingly.

Also, the sRNA for *Streptomyces coelicolor* is mislabeled. It should be Scr5239; Sco5239 is a gene encoding for a signal transduction histidine kinase in *S. coelicolor*. Please revise figure 1d accordingly.

[Response] Thank you again for thoroughly reviewing our manuscript and kindly pointing out our mistake. We have revised Figure 1d and also other parts in the manuscript and SI accordingly.

2. In line 174, the authors mentioned that the BHR-sRNA system outperformed the *E. coli* system in three of eight Gram-negative bacteria. However, in Figure 2b, this is clearly obvious for only two Gram-negative strains (*Klebsiella pneumoniae* and *Pseudomonas putida*). Which is the third strain? It would be helpful to support the comparison with statistical analysis to highlight these strains in figure 2b.

[Response] Thank you for the comment. Although not clearly visible in Figure 2b, the third strain is *E. coli* DH5 α . In this strain, employing the RoxS-BsHfq system showed a slightly higher knockdown efficiency (81.8%) than that (80.6%) by the MicC-EcHfq system. As the reviewer pointed out, such slight difference cannot be considered as significant. Thus, we added the results of statistical analyses in Figure 2b for more solid comparison of RoxS-BsHfq and MicC-EcHfq. We also listed in the revised manuscript the bacterial species showing higher knockdown efficiencies when RoxS-BsHfq was employed (Revised manuscript page 9):

“the BHR-sRNA system outperformed the MicC-EcHfq system in **seven** out of eight Gram-positive bacteria (*L. lactis*, *S. epidermidis*, *B. subtilis*, *R. opacus*, *C. xerosis*, *C. glutamicum* ATCC 13032, and *C. glutamicum* BE) and also in three out of eight Gram-negative bacteria (*P. putida*, *K. pneumoniae*, and *E. coli* DH5 α).”

3. Was the EcHfq chaperone protein tested in the Gram-positive strains codon optimised? Reduced translational efficiency of the MicC-EcHfq system in Gram-positive strains due to codon usage differences could be a confounding factor when comparing the sRNA systems. Please clarify and mention this in the text.

[Response] Thank you for the comment. The gene encoding the EcHfq chaperone tested in the Gram-positive strains was not codon optimized for each strain. Although it would have been even better if we tested codon-optimized versions as well, we did not perform codon optimization studies as we already had so many experiments to perform as can be seen in this paper. While it is true that the nonoptimal codons in the *EcHfq* gene might lead to its reduced translational efficiency in some strains, it should also be noted that codon optimization does not always lead to improved translation of a gene either. For example, in our previous study some years ago (PMID 10388699), expression of the human *obese* gene with the native nucleotide sequences resulted in higher production of human leptin when compared with that produced using the *E. coli* codon optimized gene. In our study, we have also demonstrated that the knockdown efficiency of the MicC-EcHfq system was not improved by codon optimization of

EcHfq (Supplementary Fig. 1a and 1b), at least in *C. glutamicum*. Thus, it cannot be said that the codon optimization of *BsHfq* might be a confounding factor. Also, when considering the potential application of the BHR-sRNA system in non-domesticated bacteria, it would be best to use the sRNA systems in their native sequences for ease of use. However, we agree with the reviewer that it will be worthwhile to perform codon optimization of the *BsHfq* gene as it does have the potential of improving the performance of the sRNA system. In this regard, we revised the discussion section as follows (Revised manuscript page 18):

“Such strategies include altering the promoter strength⁴⁶, codon optimization of the *BsHfq* gene, and employing plasmids with different copy numbers⁴⁷...”

4. From Figure 2b, broad-host range performance is exhibited by both *E. coli* and *B. subtilis* sRNA-Hfq systems. In some Gram-positive strains (like the *S. coelicolor*) the *E. coli* system is more efficient than the *B. subtilis* system (BHR-sRNA). Similar to the *E. coli* system (in supplementary table 6), the *B. subtilis* system has also been noted by the authors as more specific to Gram-positives (lines 176-177). The authors should be more specific in naming the BHR-sRNA system or qualify further why the *B. subtilis* system should be named BHR-sRNA and not *E. coli*.

[Response] Thank you for the comment. We have mentioned this in lines 163-169, where a repression of > 50% is achieved in a total of 12 out of 16 strains, as opposed to a total of 6 out of 16 strains using the *E. coli* system. For clarity, we added the following sentence in the revised manuscript (page 9):

“The MicC-*EcHfq* system allowed > 50% of target gene repression achieved in six bacteria (*C. glutamicum* ATCC 13032, *S. coelicolor*, *E. coli* DH5 α , *C. necator*, *V. natriegens*, and *A. hydrophila*).”

While it is true that the *E. coli* system works more efficiently in some of the strains (e.g., *S. coelicolor*), our decision to designate the *BsHfq*-RoxS sRNA system as the BHR-system was based on its capability to achieve at least >50% knockdown in a broader range of microbial host strains.

5. In lines 226-227 and supplementary figure 3d, the BHR-sRNA system reduced both cell growth and flaviolin production in *Rhodococcus opacus*. Reduced cell growth due to the introduction of BHR-sRNA could be a confounding factor for reduced flaviolin. Please provide evidence of gene knockdown by BHR-sRNA of *rppA* in *R. opacus*.

[Response] Thank you for this important comment. We compared the RppA protein expression levels in *R. opacus* using SDS-PAGE and measured the relative band intensities using Image Lab. The relative RppA protein levels were found to have decreased about 52.2% with the

introduction of the BHR-sRNA targeting *rppA*. We added these data to Supplementary Fig. 3d and made changes to the figure legend:

“d, Knockdown of *rppA* by BHR-sRNA in *R. opacus*, with measured relative band intensities of RppA from SDS-PAGE.”

Minor Comments

1. There is some incorrect labelling in Figure 2b for revision.

[Response] Thank you. We corrected the labeling in Figure 2b.

a. The x-axis for *Lactococcus lactis* should be labelled RoxS instead of RoxS-BsHfq as BsHfq was not expressed.

[Response] Thank you. We corrected the labeling accordingly in Figure 2b.

b. The x-axis for *Staphylococcus epidermidis* should be RFP and not EGFP.

[Response] Thank you. We corrected the labeling accordingly in Figure 2b.

Reviewer #3 (Remarks to the Author):

The authors have addressed the points I raised in my initial review. The side-by-side experimental comparison of BHR-sRNA and MicC-EcHfq strengthens the manuscript and will guide future users on which system to use depending on the host. The new results on the combination of CRISPRi and sRNA are a great addition to the manuscript and provide the community with a new strategy to reach higher silencing levels. I believe this system will be used by the community and I recommend the publication of this work.

[Response] Thank you very much for your generous comments.

Other changes:

In the previously submitted version of the manuscript, we miscounted the number of bacterial strains showing successful knockdown of target genes ($P < 0.017$). We thus corrected the number in the revised manuscript as follows (Revised manuscript page 9):

“The BHR-sRNA system allowed successful knockdown of the reporter genes tested (Supplementary Table 7) in 15 out of 16 bacteria tested, ...”

Reviewers' Comments:

Reviewer #2:

Remarks to the Author:

The authors have sufficiently addressed all the comments that were raised. I recommend that the manuscript be accepted for publication.

Two sets of Responses to the Reviewers' comments are enclosed as this manuscript was transferred from Nature Microbiology to Nature Communications: one for Nature Microbiology and the other for Nature Communications

Responses to Reviewers' Comments for NMICROBIOL-22010060A

Those sentences that are changed or added are shown in red in the revised manuscript and in the SI for your convenience.

Reviewer comments

Reviewer #1 Overall comment:

The authors carry out a reporter knock-down screen to identify regulatory small RNA scaffolds and RNA chaperones (Hfq homologs & orthologs) that function well in both gram-negative and gram-positive bacterial species. The authors' overall goal is to identify a more versatile approach to knocking down the expression of targeted bacterial genes, using small RNA-mediated translation repression and/or mRNA destabilization. They carried out their screen in 14 strains, including *Lactococcus lactis*, *Staphylococcus epidermidis*, *Bacillus subtilis*, *Rhodococcus opacus*, *Corynebacterium xerosis*, *Corynebacterium glutamicum*, *Staphylococcus coelicolor*, *Cupriavidus nector*, *Chromobacterium violaceum*, *Pseudomonas putida*, *Vibrio natriegens*, *Aeromonas hydrophila*, *Klebsiella pneumoniae*, and *E. coli* Nissle 1917. They found that a scaffold RNA structure from the RoxS small RNA and a Hfq homolog, both from *Bacillus subtilis*, were able to reduce reporter expression levels by 50% or more in 6 gram-positive strains and 6 gram-negative strains (out of 8 each). They then apply this knock-down system across several demonstrative applications, including [1] knocking down either *sarA* or *icaA* in a skin commensal bacterial strain (*S. epidermidis*) to reduce its biofilm formation; [2] knocking down *folA* expression in *K. pneumoniae* to reduce its multidrug resistance; [3] knocking down *rppA* or *lysA* or *pyc* in

Streptomyces griseus to alter its pigment production; and [4] knocking down several different genes (individually) in *Corynebacterium glutamicum* to increase valerolactam or methyl anthranilate production. Finally, the authors construct a library of small RNAs targeting each gene in *Corynebacterium glutamicum* to identify knock-downs that improve production of indigoidine (a blue pigment). With this last application, the authors use the information from the small RNA library to identify three beneficial targets (modified by start codon exchange) that improved fed-batch fermentation titers to 54.9 g/L (11.4% higher than the state of the art).

Overall, the work presents a more versatile approach for knocking down targeted single genes in both gram-positive and gram-negative bacteria, achieving knock-downs of about 2-fold to 10-fold. The authors demonstrate a large number of applications to show-case the breadth of strains in which the approach could be applied, including one metabolic engineering application that is progressed towards scale-up and commercialization. However, the approach as demonstrated is restricted to knocking down one gene at a time, which can limit screens to simpler phenotypes. The authors' comparisons to using CRISPR is also lop-sided and partly out-of-date.

[Response] Thank you for your time and effort of reviewing our manuscript, and providing great comments on our work. We revised the paper according to your valuable comments as detailed below. Actually, this system can be used to knock down multiple genes simultaneously. According to your comment, we have included additional results on knocking down up to three genes simultaneously for the overproduction of indigoidine as addressed in your specific comment 5.

Also, to reflect your comment on the discussions regarding CRISPR, we have revised the corresponding sections updated as follows:

(Revised manuscript page 3):

“However, the practical applications of CRISPR-based tools in bacteria are **sometimes limited due to the metabolic burden** caused by the Cas9 protein⁹.”

(Supplementary Note 2):

“This was possibly due to the increased metabolic burden by the large-sized dCas9 protein (~160 kDa) ... In addition, the transcript levels of GFP were retained when using the BHR-sRNA system while 35.4 % decrease in GFP transcript level was observed using the CRISPRi system (P = 0.047; **Supplementary Fig. 2c**), indicating that sRNA represses genes at the translational level unlike CRISPRi which represses genes at the transcriptional level.”

We also added Supplementary Table 6 to provide a fair comparison between different target gene knockdown systems.

Specific Comments:

1. What is the homology (amino acid similarities) between *E. coli* Hfq and *B. subtilis* Hfq? What are the sequence & structural similarities between the *E. coli* sgRNA micC domain and the sgRNA roxS domain? Why do these differences provide an improved knock-down effects in both gram-positive & gram-negative strains as compared to *E. coli* micC and Hfq?

[Response] Thank you for the comment. The homology between *E. coli* Hfq and *B. subtilis* Hfq is 45%. For more explicit comparison of the homology of the Hfq proteins, we also replaced the nucleotide sequences of the *hfq* genes into the amino acid sequences of the Hfq proteins in Supplementary Table 5. On the other hand, there is no sequence similarity between *E. coli* MicC and *B. subtilis* RoxS. In addition, there is no sequence similarity among all the scaffold sequences used in this study. Nevertheless, the predicted structures of the RNA scaffolds (hairpin structure) are similar, which is now provided in Supplementary Fig. 4. We speculate that the higher GC content of the RoxS scaffold (51.4%) compared to that of MicC (43%) might have affected the scaffold stability and thus improved knockdown effect observed in both Gram-positive and Gram-negative strains. However, the exact mechanism is still unclear.

We added such discussion in the revised manuscript as follows (Revised manuscript page 9):

“The generally improved knockdown effect of the BHR-sRNA system in Gram-positive bacteria might be explained by the evolutionary location of *B. subtilis* closer to many of the Gram-positive bacteria (Fig. 2a). In addition, the GC content of the RoxS scaffold (51.4%) is higher than that (43%) of the previously developed *E. coli* MicC scaffold which might have affected the scaffold stability, thus improving the knockdown effect in some Gram-negative bacteria (Supplementary Fig. 4). Further studies will be needed to understand the exact mechanisms affecting the knockdown efficiencies.”

2. What are the effects of expressing *B. subtilis* Hfq in gram positive strains when using non-rich & defined medias (e.g. MOPS, M9 + various sugar)? How much does the specific growth rate change when using more operationally relevant medias?

[Response] Thank you for the comment. As it is difficult to grow many of the Gram-positive strains listed here on non-rich and defined media in the first place (due to the lack of characterization of the strains for defined media), we tested the effect of *B. subtilis* Hfq (BsHfq) in *C. glutamicum*, a representative Gram-positive bacterium. The growth profile of *C. glutamicum* harboring BsHfq in the CGXII medium, a chemically defined medium, was almost identical with that of *C. glutamicum* harboring the empty pEKEx1 plasmid (Supplementary Fig. 2b). We thus added the following sentence in the revised manuscript (Revised manuscript page 6):

“We also observed that neither BsHfq nor the BHR-sRNA system influenced the growth profile of *C. glutamicum* when grown in a chemically defined CGXII medium (Supplementary Fig. 2b).”

3. Regarding the statement, “The highest indigoidine titer (0.515 g/L) could be achieved by *ruvA* knockdown, which corresponds to 3.6-fold increase compared with that (0.144 g/L) obtained with the control strain ($P = 3.5 \times 10^{-5}$ 260 ; Fig. 261 3f). Interestingly, the five knockdown targets identified are not involved in the mainstream metabolic pathway for indigoidine production, but are rather involved in membrane components (*NCgl0549*, *NCgl2907*), genetic recombination and DNA repair (*ruvA*), IMP production (*purC*), or is hypothetical (*NCgl2113*)”.

What were the specific growth rates for these strains? It’s possible (likely) that knocking down these non-metabolic proteins will reduce the cells’ specific growth rate, which could indirectly increase production titers without a corresponding increase in productivity. What is the specific productivities for indigoidine with vs. without these gene knockdowns?

[Response] Thank you for the great comment. As the reviewer mentioned, the tradeoff between cell growth and production level exists. Finding a point of balance is important, and we tried to achieve this by high-throughput colorimetric screening – finding suitably growing strains with the highest production titers. Of course, finding engineering targets that would promote the production of target chemicals without compromising cell growth would be the best option.

To see the effects of knocking down the gene targets (*NCgl2907*, *NCgl2113*, and *ruvA*) applied to the final indigoidine producer, we calculated the specific growth rates and the productivities of the BIRU11 strain (the base strain) and the BIRU20 strain (the base strain knocked down with *NCgl2907*, *NCgl2113*, and *ruvA*) by fed-batch fermentations. The maximum specific growth rate of the BIRU11 strain was $3.7 \text{ g l}^{-1} \text{ h}^{-1}$ (Ghiffary, M.R. et al. ACS Sustain. Chem. Eng. 9, 6613-6622 (2021)) while that of the BIRU20 strain was $1.9 \text{ g l}^{-1} \text{ h}^{-1}$ (Fig. 3h). The maximum productivity of the BIRU11 strain was $2.3 \text{ g l}^{-1} \text{ h}^{-1}$ while that of the BIRU20 strain was $2.8 \text{ g l}^{-1} \text{ h}^{-1}$. We also calculated the molar yields [$\text{mol}(\text{indigoidine}) \text{ mol}(\text{consumed_glucose})^{-1}$] and the contents [$\text{g}(\text{indigoidine}) \text{ gDCW}^{-1}$] of the two strains, and provided the data in Supplementary Table 11, together with other production metrics. As shown in the table below, while the maximum specific growth rate of the BIRU20 strain was lower than that of the BIRU11 strain, the titer, productivity, yield, and content obtained with the BIRU20 strain were all increased.

Supplementary Table 11 | Comparison of growth rate and indigoidine production during the fed-batch cultures of BIRU11 and BIRU20 strains.

Strain	BIRU11 ^a	BIRU20 ^b
Titer (g l ⁻¹)	49.3	54.9
Productivity (g l ⁻¹ h ⁻¹)	0.967	1.04
Content (g g ⁻¹) ^c	0.758	0.840
Yield (mol mol ⁻¹) ^d	0.1	0.181

Maximum specific growth rate (g l ⁻¹ h ⁻¹)	3.70	1.91
Maximum specific productivity (g l ⁻¹ h ⁻¹)	2.30	2.77

^a Data from Fig. 4a of Ghiffary, M.R. et al. ACS Sustain. Chem. Eng. 9, 6613-6622 (2021).

^b Data from Fig. 3h

^c g(indigoidine) gDCW⁻¹

^d mol(indigoidine) mol(consumed glucose)⁻¹

We also added the following sentence in the revised manuscript (Revised manuscript page 16):

“The productivity, content, and yield were also all increased with the tradeoff of decreased maximum specific growth rate (Supplementary Table 11).”

4. RuvA is a DNA helicase responsible for accelerating genomic recombination. The authors don't explain why knocking down RuvA could improve metabolic productivity, but the most likely mechanism is that their engineered genetic system contains repetitive DNA that is triggering homologous recombination, causing self-deletion. Knocking down RuvA could be slowing down this process. The authors need to identify a plausible mechanism to explain this knock-down effect, for example, by sequencing isolates from post-fermentation runs and identifying/counting the frequency of mutations that appear with vs. without knocking down RuvA.

[Response] Thank you for the great comment and your insight regarding *ruvA* knockdown. Due to the intricate nature of biological systems, it is sometimes very difficult to interpret the effect of knocking down a non-obvious target gene on the production level of a target chemical. We have also observed that the knockdown of *ytfR* (encoding sugar ABC transporter ATPase), which is not directly linked to chemical production, resulted in significant enhancement of violacein production (Yang et al., Metab Eng 2019, PMID 30999052). As the reviewer suggested, it seems plausible that knocking down RuvA altered the host homologous recombination system, and thereby bringing less mutations in the production host. However, as complete elucidation of this mechanism is beyond the scope of this study, we did not perform further experiments on sequencing the genomes of the cells obtained after fermentations. Nevertheless, we additionally discussed on this in the revised manuscript as the reviewer kindly commented with great insight (Revised manuscript page 16):

“Since RuvA is a DNA helicase responsible for accelerating genomic recombination, knockdown of *ruvA* would have resulted in improved genetic stability of the indigoidine-producing strain due to the reduced homologous recombination. However, the exact mechanism on how knocking down *ruvA* resulted in improved indigoidine production requires further study.”

5. The authors do not mention any usage of their approach to knocking down the expression of multiple genes even though this capability is highly desired. The authors should use their system to knock-down the expression of two genes at the same time or very clearly explain why that is not possible using their system. Prior approaches exist that use CRISPR to knock-down the expression of multiple bacterial genes [Reis, A. C., Halper, S. M., Vezeau, G. E., Cetnar, D. P., Hossain, A., Clauer, P. R., & Salis, H. M. (2019). Simultaneous repression of multiple bacterial genes using nonrepetitive extra-long sgRNA arrays. *Nature biotechnology*, 37(11), 1294-1301.]

[Response] Thank you for the critical comment. To address this great comment, we performed additional experiments for the simultaneous knockdown of the reporter genes encoding mCherry and GFP using the BHR-sRNA platform. We also observed orthogonal knockdown of either reporter by the introduction of either sRNA into the dual reporter system. Thus, the following sentences were added to the revised manuscript (Revised manuscript page 6):

“Next, the BHR-sRNA system was tested for the simultaneous knockdown of multiple target genes. Employing sfGFP and mCherry as the dual fluorescent reporters, simultaneous knockdown of the both reporters was achieved by the introduction of a single plasmid harboring the anti-sfGFP and the anti-mCherry sRNAs (Supplementary Fig. 1e). Orthogonal knockdown of either reporter was also achieved by the introduction of each sRNA to the *C. glutamicum* strain harboring the dual reporters.”

For actual metabolic engineering applications, we also tested the simultaneous knockdown of two and three genes (selected among the top three genes *NCgl2907*, *ruvA*, or *NCgl2113*) for the production of indigoidine, and observed that the results were in good correlation with those obtained by the genome engineered (codon exchange) strains in Figure 3g. We added these data to the Supplementary Fig. 7e. Please note that the data in Figure 3g were obtained by flask culture (engineered from the BIRU-NP strain) while the data in Figure 3f and Supplementary Fig. 7e were obtained by test tube-scale culture (engineered from the WT-BpsA strain), leading to discrepancies in the titers. We also added the corresponding discussion in the revised manuscript as follows (Revised manuscript page 16):

“We also tested the combinatorial knockdown of the top three (*NCgl2113*, *ruvA*, and *NCgl0549*) targets by introduction of plasmids harboring two or three sRNAs (Supplementary Fig. 7e). Although simultaneous knockdown of all the three genes did not lead to the highest production, three out of four sRNA combinations resulted in higher production of indigoidine when compared with that by the *ruvA*-knockdown strain (Supplementary Fig. 7e).”

Reviewer #2 Overall comment:

In this study, the authors developed and demonstrated the use of a broad-host range synthetic sRNA (bhr-sRNA) platform comprising a sRNA scaffold and Hfq chaperone from *Bacillus subtilis*. Utilising the bhr-sRNA platform, efficient gene knockdowns in various bacteria strains of industrial and therapeutic interest were established. Collectively, the capabilities of the bhr-sRNA platform were demonstrated, which will advance the engineering potential of different bacteria strains of industrial and medical interest. Below are some questions that require clarification.

[Response] Thank you for your time and effort of reviewing our manuscript, and providing great comments on our work. We revised our paper according to your valuable comments as detailed below.

1. In Figure 1, it is shown that GFP fluorescence was reduced significantly even in the absence of chaperone expression (Bshfq/RoxS, Sahfq/Spr, Arn), especially Bshfq/RoxS that was selected for the bhr-sRNA platform. Is it possible that RoxS sRNA scaffold alone can or more efficiently knockdown gene expression without Bshfq? It also seems that the knockdown of *upp* in *Lactococcus lactis* did not involve the expression of Bshfq (see question 2a). Please explain and substantiate the final design of bhr-sRNA needing the chaperone with the sRNA scaffold.

[Response] Thank you for the comment. According to our data in Figure 1, it is true that RoxS itself allows significant knockdown of a target gene. However, the additional introduction of BsHfq further increased the knockdown efficiency which was statistically significant ($P = 0.012$). To clarify this, we added the following sentence in the revised manuscript to describe this (Revised manuscript page 6):

“The scaffold RoxS only also resulted in 58.0% knockdown of *GFP*, but the knockdown efficiency was lower than that (64.6%) by the RoxS-BsHfq system ($P = 0.012$).”

In the case of *L. lactis*, please refer to the response to the question 2a.

2. The following questions pertain to Figure 2.

a. For Figure 2b, it seems that only RoxS was expressed for the knockdown of *upp* in *Lactococcus lactis*. What is the reason for not having Bshfq expressed in *L. lactis*?

[Response] Thank you for the comment. The reason for not having BsHfq in *L. lactis* was because we had failed to clone the plasmid with *hfq* to target the *upp* gene in *L. lactis*, possibly due to the toxicity. However, as we have demonstrated earlier that using the RoxS scaffold is also capable of significant knockdown, we decided to use only RoxS for *L. lactis*. To clarify this, we have added the following sentence in the manuscript (Revised manuscript page 9):

“For testing in *L. lactis*, only RoxS was used to knockdown the *upp* gene as the construction of the sRNA plasmid harboring *BsHfq* was unsuccessful.”

b. In the manuscript, it was mentioned that bhr-sRNA platform was tested in seven strains of gram-negative species (line 117). It was later mentioned in line 144 that there are 8 gram-negative bacteria (presumably *Escherichia coli* DH5a was included). However, in figure 2b, the results for MicC-EcHfq and not RoxS-BsHfq was shown for *E. coli* DH5a. Was RoxS-BsHfq tested for *E. coli* DH5a? Please clarify this discrepancy.

[Response] Thank you for pointing this out. We apologize for the mistake. The data shown in Figure 2b for *E. coli* DH5a corresponds to the results for RoxS-BsHfq. We revised Figure 2b accordingly. To clarify this, we revised the manuscript as follows (Revised manuscript page 7, 9):

“Seven Gram-positive bacterial species (in addition to *C. glutamicum* ATCC 13032) corresponding to the phyla of...”

“and seven Gram-negative bacterial species (in addition to *E. coli* DH5a) corresponding to the classes of”

“The BHR-sRNA system allowed successful knockdown of the reporter genes tested (Supplementary Table 7) in 14 out of 16 bacteria tested...”

c. The statistical significance for Figure 2d seems to be absent. Please include it.

[Response] We added the statistical significance to Figure 2d as suggested.

3. Altering promoter strength and employing plasmids of different copy numbers (lines 148-149) seem relatively generic in terms of optimisation strategies. For example, the bhr-sRNA platform does not seem to have an effect in *Chromobacterium violaceum*. What are the possible limitations of bhr-sRNA and would the mentioned optimisation strategies be sufficient to address these limitations? Please elaborate in greater detail.

[Response] Thank you for the comment. We further elaborated this aspect in the discussion section in the revised manuscript as follows (Revised manuscript page 18):

“For those bacteria showing relatively lower knockdown efficiencies (Fig. 2b), further strain-dependent optimization of the sRNA platform will enable more efficient knockdown. Such strategies include altering the promoter strength⁴⁶ and employing plasmids with different copy numbers⁴⁷ that have been previously demonstrated in *E. coli*. In addition, the BHR-sRNA can be

employed together with CRISPRi for the dual transcriptional and translational repression of target genes, which was shown to result in more efficient knockdown, albeit at the cost of reduced cell growth (Supplementary Fig. 2). Another potential strategy is engineering the Hfq protein, as Hfq is known to aid the binding of sRNA to the target mRNA⁴⁸.”

4. In Extended Data 2c, as flaviolin is a red pigment, does OD600 interfere with flaviolin absorbance and emission? If so, would the results for *Rhodococcus opacus* be significant? Please indicate if normalised OD600 for *R. opacus* data were used in this figure.

[Response] For measuring the OD₆₀₀ values, we centrifuged the cells and resuspended them in PBS to eliminate the interference of flaviolin on the measurement of cell growth. To clarify this, we revised the methods section in the revised manuscript as follows (Revised manuscript page 24):

“When measuring the OD₆₀₀ values of the *R. opacus* and *E. coli* strains harboring *rppA*, the cultures were centrifuged and the pellets were resuspended in equal volumes of PBS to eliminate the interference of flaviolin on the OD₆₀₀ measurement for cell growth.”

As the reviewer pointed, however, the reduced flaviolin production from the sRNA-harboring strain might be partly due to the reduced cell growth. Thus, we revised the corresponding sentence as follows (Revised manuscript page 11):

“Knockdown of *rppA* by the BHR-sRNA system in *R. opacus* resulted in reduced flaviolin production as well as reduced cell growth (Supplementary Fig. 3d).”

5. In the abstract, line 25 mentions efficient target gene knockdown in 16 bacterial species. It would be more accurate to describe that the bhr-sRNA platform was tested in 16 strains with more than 50% efficiency of gene knockdown for X number of strains.

[Response] Thank you for the suggestion. We revised the abstract accordingly as follows:

“BHR-sRNA was tested in 16 bacterial species including commensal, probiotic, pathogenic, and industrial bacteria, with > 50% of target gene knockdown achieved in 12 bacterial species.”

Reviewer #3 Overall comment:

The authors developed a new sRNA-based system to knockdown the expression of target genes in a large diversity of microorganisms. Confronted with the host-range limitations of their previous system which was initially developed for *E. coli* (Na et al. 2013), the authors decided to systematically test the performance of natural sRNA systems from various Gram positive bacteria in a gram positive and industrially relevant host: *C. glutamicum*. After selecting the system from *B. subtilis*, they tested its performance in *E. coli* DH5alpha and on a large panel of 14 phylogenetically diverse bacteria. The authors later provide several examples of applications of their system (virulence factor repression in pathogenic bacteria and metabolic engineering for enhanced production of bulk and fine chemicals in model industrial strains).

Synthetic biology tools to enable facile modification of expression levels represent a clear need in the community, as exemplified by the popularity of the original sRNA system developed by the authors (Na et al. 2013). Host-range considerations are also key to enable real world applications and a simple, well-characterized, broad-host-range silencing tool will most certainly be a contribution highly appreciated by the community. This work is very timely and I congratulate the authors on this project, in particular considering the amount of work it represents. Among the examples of applications provided by the authors, the sRNA-mediated targeting of all 2,959 genes of *C. glutamicum* and the discovery of non-evident targets improving indigoidine production to the highest titer reported to date is particularly elegant and will certainly inspire several metabolic engineering projects in the future.

While adoption of this tool to some degree is almost guaranteed in my opinion on the basis of the results presented here, most labs faced with a need for programmable knock-down capabilities will read this paper looking for pros and cons while hesitating between the multiple tools available to date. The authors could significantly increase the rapid adoption of this method by providing additional elements to guide the choice between CRISPRi, MicC-EcHfq or RoxS-BsHfq.

[Response] Thank you for your time and effort of reviewing our manuscript, and providing great comments on our work. We revised the paper according to your valuable comments as detailed below. For the comment regarding the discussion on CRISPRi, MicC-EcHfq, and RoxS-BsHfq, please refer to the response to this reviewer's first minor point below.

Specific comments:

-Given the claim on broad-host-range and the vast amount of work represented by the development of protocols to characterize knock-down effects in 14 bacteria (various vectors, transformation methods, culture conditions, reporter genes) I regret that this know-how of the authors is not more leveraged in this work to compare BHR-sRNA to MicC-EcHfq and to CRISPRi in more species (instead of only one gram positive bacteria, *C. glutamicum*). It seems that given the protocols

established by the authors this would mostly require limited cloning. Regardless of whether or not BHR-sRNA is indeed systematically performing better as the name suggests, such a dataset would be of great value to guide the decision of users working with a particular bacteria when choosing which tool is most adequate in their case.

[Response] Thank you for the great suggestion. We have performed additional experiments to compare the knockdown efficiencies by the BHR-sRNA system and the MicC-EcHfq system in diverse bacteria. We thus provided the data in Figure 2b. We added the corresponding discussion in the revised manuscript as follows (Revised manuscript page 9):

“We also tested the knockdown of the above reporter genes in 16 bacteria by employing the MicC-EcHfq system, and found that the BHR-sRNA system outperformed the MicC-EcHfq system in six out of eight Gram-positive bacteria and also in three out of eight Gram-negative bacteria.”

On the other hand, we did not test the CRISPRi system in different bacteria since cloning of the large-sized dCas9 genes to many different plasmid platforms and direct comparison of the sRNA system with the CRISPRi system are beyond the scope of this study.

-The authors' claim that “altering the promoter strength and employing plasmids with different copy numbers will enable more efficient knockdown.” appears too assertive and not obvious without more evidence. The authors should provide evidence for this or reformulate the sentence. An experimental example of such strain-dependent finetuning would provide readers with very valuable information on the possibility to troubleshoot BHR-sRNA silencing efficiencies. More generally the lack of information on the impact of the expression level of Hfq and the sRNA itself on the silencing efficiency is one of the weak points of the study from a synthetic biology perspective.

[Response] Thank you for the great comment. We revised this section and moved to the discussion as follows (Revised manuscript page 18):

“For those bacteria showing relatively lower knockdown efficiencies (Fig. 2b), further strain-dependent optimization of the sRNA platform will enable more efficient knockdown. Such strategies include altering the promoter strength⁴⁶ and employing plasmids with different copy numbers⁴⁷ that have been previously demonstrated in *E. coli*. In addition, the BHR-sRNA can be employed together with CRISPRi for the dual transcriptional and translational repression of target genes, which was shown to result in more efficient knockdown, albeit at the cost of reduced cell growth (Supplementary Fig. 2). Another potential strategy is engineering the Hfq protein, as Hfq is known to aid the binding of sRNA to the target mRNA⁴⁸.”

-I find it surprising that despite a 23X coverage library and 68,700 colonies screened the 108 colonies selected appear to have mostly (if not only) different targets reported in Extended Data Fig. 5a. Intuitively, such an experiment would lead to selecting several times the same target (23 times in theory), I understand that there is colony variability but is this variability so high that the experiment appears to lead to selecting targets almost at random ? Because this approach could inspire a lot of other labs, the authors should address this unexpected result in the main text and if no explanation is found they must acknowledge the limits of such systematic screening in the case of indigoidine to warn the readers considering to attempt a similar screening with BHR-sRNA.

[Response] Thank you for the excellent comment. As suggested, we discussed the screening procedure and the result in more detail in the revised manuscript as follows (Revised manuscript page 14):

“Among 108 initially screened colonies, there were duplicate colonies for six knockdown gene targets (*NCgl109*, *NCgl574*, *NCgl755*, *NCgl1496*, *NCgl1540*, and *NCgl2427*), and triplicate colonies for one knockdown gene target (*NCgl1893*) (Supplementary Fig. 7a). However, some of these colonies harboring identical sRNAs showed different indigoidine production levels, indicating colony variations in the initial screening stage.”

Minor points:

-Because the manuscript refers several times to the limitations of CRISPRi the authors should acknowledge recent claims of improvements with the CRISPRi method (such as <https://doi.org/10.1093/nar/gkaa294>) and ideally provide a fair comparison in the form of a table.

[Response] Thank you for the comment. We have revised the corresponding sections as the reviewer suggested:

(Revised manuscript page 3):

“However, the practical applications of CRISPR-based tools in bacteria are sometimes limited due to the metabolic burden caused by the Cas9 protein⁹.”

(Supplementary Note 2):

“This was possibly due to the increased metabolic burden by the large-sized dCas9 protein (~160 kDa) ... In addition, the transcript levels of GFP were retained when using the BHR-sRNA system while 35.4 % decrease in GFP transcript level was observed using the CRISPRi system (P = 0.047; Supplementary Fig. 2c), indicating that sRNA represses genes at the translational level unlike CRISPRi which represses genes at the transcriptional level.”

Also, as the reviewer suggested, we added Supplementary Table 6 which compares the BHR-sRNA system with CRISPRi and the MicC-EcHfq system.

-The knockdown efficiencies reported by the authors for CRISPRi and BHR-sRNA system in *C. glutamicum* are surprisingly similar (65.4% and 65.2%, respectively). Given that the mechanism of silencing is completely different such close values are a bit unexpected. It would be great to see whether this silencing efficiency can be further increased by using a combination of both systems. This would show whether this almost identical efficiency is purely fortuitous or if another reason is limiting the maximum silencing efficiency. An example leveraging the effect of both transcriptional and translational level silencing would also have the advantage to present this method as potentially synergistic with CRISPRi for applications requiring strong silencing rather than only as a competing approach.

[Response] Thank you for this great suggestion. We performed additional experiments for the simultaneous knockdown experiments combining both CRISPRi and BHR-sRNA systems to knockdown GFP. Indeed, the maximum silencing efficiency was increased when we combined both CRISPRi and BHR-sRNA (as the reviewer nicely commented), most likely due to the combined effect of silencing at both transcriptional and translational levels. However, decreased cell growth was observed when both BHR-sRNA and CRISPRi were employed. We included these results in Supplementary Fig. 2d,e and made the following changes to our revised manuscript (Revised manuscript page 7):

“In addition, we tested whether employing the BHR-sRNA system together with CRISPRi can further enhance knockdown efficiencies by the dual repression at both transcriptional and translational levels. Although the knockdown efficiency by the dual BHR-sRNA/CRISPRi system (78.5%) was higher than that (64.6%) by the BHR-sRNA system (Supplementary Fig. 2d), the dual BHR-sRNA/CRISPRi system was not chosen due to the decreased cell growth and the complexity of the system (Supplementary Fig. 2e). Comparison of the BHR-sRNA system with CRISPRi and the MicC-EcHfq system is summarized in Supplementary Table 6.”

-The authors claim that “The BHR-sRNA system allowed successful knockdown of the reporter genes tested (Supplementary Table 6) in all bacteria” should be reformulated and nuanced by the fact that they report only minor differences in *C. violaceum* (The authors labelled this data as “Not Significant”) and it is unclear whether the lower fluorescence signal observed in *R. opacus* is due to lower cell density observed in this experiment or due to the silencing (The EGFP/OD600 ratio is in fact increased rather than decreased). The BHR-sRNA does appear functional in *R. opacus* when applied to knocking down RppA.

[Response] Thank you for pointing this out. We revised the corresponding sentence as follows (Revised manuscript page 8):

“The BHR-sRNA system allowed successful knockdown of the reporter genes tested (Supplementary Table 6) in 14 out of 16 bacteria tested, with > 50% of target gene repression achieved...”

In the case of *rppA* knockdown test in *R. opacus*, the result is also not very clear due to the reduced growth upon introduction of the sRNA. Thus, we also revised the corresponding sentence as follows (Revised manuscript page 11):

“Knockdown of *rppA* by the BHR-sRNA system in *R. opacus* resulted in reduced flavin production as well as reduced cell growth (Supplementary Fig. 3d).”

-In general, because this work involves many bacteria it would be clearer to have every graph labelled on top with the corresponding bacteria, as is done in Figure 2.

[Response] Thank you for the nice suggestion. We revised our Figures (Figure 2 and Supplementary Fig. 3) to accommodate your suggestion.

-Fig 2b: The figure for E coli Dh5a indicates that MicC is used, is this mislabeled ?

[Response] Thank you for pointing this out. We apologize for the mistake. We used the BsHfq-RoxS system and corrected Fig. 2b accordingly.

-To harmonize with the other vignettes, the MANT production vignette should report production titer numbers in the main text. In general the MANT vignette is not very convincing, in particular it is unclear why in Fig4c CT produces 45mg/L and in Fig4d CT produces around 190mg/L. Given the small differences observed with the knockdown strains this contributes in bringing some doubt on the effect. The authors should address this point in the text.

[Response] Thank you for the suggestion. The CT strain in Supplementary Fig. 6c has all the MANT producing genes in one plasmid and the BHR-sRNA system in another, while the CT strain in Supplementary Fig. 6e has the MANT producing genes being expressed from two plasmids where the configurations have been optimized previously for MANT production. For clarity, we revised the sentence as follows (Revised manuscript page 12):

“First, we validated three target genes (*gnd*, *tkl*, and *pgl*) that improved MANT production titers in *C. glutamicum* harboring MANT biosynthetic genes in one plasmid and the BHR-sRNA system in another plasmid (Supplementary Fig. 6c; Supplementary Table 9, Supplementary Note 4). ... Shake flask culture of the resultant strains demonstrated 16% increase in the MANT production titers from 192 to 223 mg l⁻¹ in the final engineered *C. glutamicum* strain in which *gnd* was knocked down (harboring two plasmids containing genes for MANT production; Supplementary Fig. 6e).”

Detailed description on the production strains can be found in Supplementary Note 4.

-Line 220: “these results demonstrate not only the first production of a lactam in *C. glutamicum* from glucose, but also valerolactam production to the highest level.” The authors should be more specific, is this the highest level reported so far?

[Response] Yes, it is the highest valerolactam production through biological means reported to date.

We revised the sentence as follows (Revised manuscript page 13):

“Taken together, these results demonstrate not only the first production of a lactam in *C. glutamicum* from glucose, but also **the highest valerolactam production reported.**”

-Line 226: “Here, we not only validated three target genes that improved MANT production titers (Extended Data Fig. 4c), but also demonstrated 16% increase in MANT production titers without the use of BHR-sRNA plasmids through genome engineering (Extended Data Fig. 4d).”

The authors should explain their genome engineering strategy (ATG to GTG) here in the main text, instead of doing it only later for Indigoidine.

[Response] Thank you. We revised the section as suggested (Revised manuscript page 13):

“Next, we examined whether the beneficial effects of gene knockdown on MANT production can be translated into genome engineering so that sRNA plasmid-free strains can be developed. The three targets identified above were engineered by changing the start codon of the chromosomal target genes from ATG to GTG or TTG to endow gene knockdown effect (Supplementary Fig. 6d).”

-Line 479: What is Supplementary Fig. 17 referring to ?

[Response] Sorry for the mistake. It is Supplementary Fig. 7f and is now corrected.

-Supplementary Figure 1 is not referred to in the manuscript

[Response] Thank you for pointing this out. We corrected (Revised manuscript page 20):

“...installed at the KAIST Bio Core Center was also used through a 530/30 band-pass filter for the GFP emission spectrum (Fig. 1d, **Supplementary Fig. 2a**; **Supplementary Fig. 8**).”

-The authors should mention in the main text whether all the bacteria on which BHR-sRNA was tested so far are reported here, or if some attempts were inconclusive and excluded from the results. The information of failed attempts, if any, would be highly valuable for users considering to use BHR-sRNA in bacteria that are not listed here.

[Response] Thank you for the great comment. We also attempted to test the BHR-sRNA system in another bacterium (*Lactobacillus acidophilus*). However, we did not add the data not because the results were bad but because we could not develop basic engineering protocols for this bacterium (e.g., transformation). Except for this strain case, we did not exclude any data and showed the results of all the bacterial strains we tested.

Responses to Reviewers' Comments for NCOMMS-23-03181A

Those sentences that are changed or added are shown in red in the revised manuscript and in the SI for your convenience.

REVIEWER COMMENTS

Reviewer #1 (Remarks to the Author):

The authors have responded to and addressed all of this reviewer's questions.

[Response] Thank you very much.

Reviewer #2 (Remarks to the Author):

In this work, the authors first developed a synthetic sRNA scaffold platform comprising the sRNA scaffold and Hfq chaperone from *Bacillus subtilis* in *Corynebacterium glutamicum*. The authors then demonstrated the use of this platform in 16 bacterial strains and compared its efficiency to the sRNA platform developed from *Escherichia coli*. Lastly, the authors showed how the platform could be applied for virulence gene knock-down and genome-scale candidate identification in metabolic engineering for enhanced chemical production.

In general, the manuscript is well-written and easy to understand. This study addresses a technological gap for the quick experimental validation of target genes, which metabolic engineering needs. Below are my comments which require clarification before publication.

Major Comments

1. In lines 96-97, it was mentioned that all corresponding Hfq proteins were introduced with their corresponding sRNA. However, in Figure 1d, *Bacillus thuringiensis* sRNA Bts was introduced with *B. subtilis* Hfq protein instead. Please revise Figure 1d and provide data reflective of the *Bacillus thuringiensis* sRNA system.

[Response] Sorry for the mistake and thank you for finding this error. This was an incorrect labelling. *Bacillus thuringiensis* sRNA BtsR1 was in fact introduced with *B.thuringiensis* Hfq (Supplementary Table 2). We have revised Figure 1d accordingly.

Also, the sRNA for *Streptomyces coelicolor* is mislabeled. It should be Scr5239; Sco5239 is a gene encoding for a signal transduction histidine kinase in *S. coelicolor*. Please revise figure 1d accordingly.

[Response] Thank you again for thoroughly reviewing our manuscript and kindly pointing out our mistake. We have revised Figure 1d and also other parts in the manuscript and SI accordingly.

2. In line 174, the authors mentioned that the BHR-sRNA system outperformed the *E. coli* system in three of eight Gram-negative bacteria. However, in Figure 2b, this is clearly obvious for only two Gram-negative strains (*Klebsiella pneumoniae* and *Pseudomonas putida*). Which is the third strain? It would be helpful to support the comparison with statistical analysis to highlight these strains in figure 2b.

[Response] Thank you for the comment. Although not clearly visible in Figure 2b, the third strain is *E. coli* DH5 α . In this strain, employing the RoxS-BsHfq system showed a slightly higher knockdown efficiency (81.8%) than that (80.6%) by the MicC-EcHfq system. As the reviewer pointed out, such slight difference cannot be considered as significant. Thus, we added the results of statistical analyses in Figure 2b for more solid comparison of RoxS-BsHfq and MicC-EcHfq. We also listed in the revised manuscript the bacterial species showing higher knockdown efficiencies when RoxS-BsHfq was employed (Revised manuscript page 9):

“the BHR-sRNA system outperformed the MicC-EcHfq system in **seven** out of eight Gram-positive bacteria (*L. lactis*, *S. epidermidis*, *B. subtilis*, *R. opacus*, *C. xerosis*, *C. glutamicum* ATCC 13032, and *C. glutamicum* BE) and also in three out of eight Gram-negative bacteria (*P. putida*, *K. pneumoniae*, and *E. coli* DH5 α).”

3. Was the EcHfq chaperone protein tested in the Gram-positive strains codon optimised? Reduced translational efficiency of the MicC-EcHfq system in Gram-positive strains due to codon usage differences could be a confounding factor when comparing the sRNA systems. Please clarify and mention this in the text.

[Response] Thank you for the comment. The gene encoding the EcHfq chaperone tested in the Gram-positive strains was not codon optimized for each strain. Although it would have been even better if we tested codon-optimized versions as well, we did not perform codon optimization studies as we already had so many experiments to perform as can be seen in this paper. While it is true that the nonoptimal codons in the *EcHfq* gene might lead to its reduced translational efficiency in some strains, it should also be noted that codon optimization does not always lead to improved translation of a gene either. For example, in our previous study some years ago (PMID 10388699), expression of the human *obese* gene with the native nucleotide sequences resulted in higher production of human leptin when compared with that produced using the *E. coli* codon optimized gene. In our study, we have also demonstrated that the knockdown efficiency of the MicC-EcHfq system was not improved by codon optimization of *EcHfq* (Supplementary Fig. 1a and 1b), at least in *C. glutamicum*. Thus, it cannot be said that the codon optimization of *BsHfq* might be a confounding factor. Also, when considering the

potential application of the BHR-sRNA system in non-domesticated bacteria, it would be best to use the sRNA systems in their native sequences for ease of use. However, we agree with the reviewer that it will be worthwhile to perform codon optimization of the *BsHfq* gene as it does have the potential of improving the performance of the sRNA system. In this regard, we revised the discussion section as follows (Revised manuscript page 18):

“Such strategies include altering the promoter strength⁴⁶, codon optimization of the *BsHfq* gene, and employing plasmids with different copy numbers⁴⁷...”

4. From Figure 2b, broad-host range performance is exhibited by both *E. coli* and *B. subtilis* sRNA-Hfq systems. In some Gram-positive strains (like the *S. coelicolor*) the *E. coli* system is more efficient than the *B. subtilis* system (BHR-sRNA). Similar to the *E. coli* system (in supplementary table 6), the *B. subtilis* system has also been noted by the authors as more specific to Gram-positives (lines 176-177). The authors should be more specific in naming the BHR-sRNA system or qualify further why the *B. subtilis* system should be named BHR-sRNA and not *E. coli*.

[Response] Thank you for the comment. We have mentioned this in lines 163-169, where a repression of > 50% is achieved in a total of 12 out of 16 strains, as opposed to a total of 6 out of 16 strains using the *E. coli* system. For clarity, we added the following sentence in the revised manuscript (page 9):

“The MicC-EcHfq system allowed > 50% of target gene repression achieved in six bacteria (*C. glutamicum* ATCC 13032, *S. coelicolor*, *E. coli* DH5 α , *C. necator*, *V. natriegens*, and *A. hydrophila*).”

While it is true that the *E. coli* system works more efficiently in some of the strains (e.g., *S. coelicolor*), our decision to designate the BsHfq-RoxS sRNA system as the BHR-system was based on its capability to achieve at least >50% knockdown in a broader range of microbial host strains.

5. In lines 226-227 and supplementary figure 3d, the BHR-sRNA system reduced both cell growth and flaviolin production in *Rhodococcus opacus*. Reduced cell growth due to the introduction of BHR-sRNA could be a confounding factor for reduced flaviolin. Please provide evidence of gene knockdown by BHR-sRNA of *rppA* in *R. opacus*.

[Response] Thank you for this important comment. We compared the RppA protein expression levels in *R. opacus* using SDS-PAGE and measured the relative band intensities using Image Lab. The relative RppA protein levels were found to have decreased about 52.2% with the introduction of the BHR-sRNA targeting *rppA*. We added these data to Supplementary Fig. 3d and made changes to the figure legend:

“d, Knockdown of *rppA* by BHR-sRNA in *R. opacus*, with measured relative band intensities of RppA from SDS-PAGE.”

Minor Comments

1. There is some incorrect labelling in Figure 2b for revision.

[Response] Thank you. We corrected the labeling in Figure 2b.

a. The x-axis for *Lactococcus lactis* should be labelled RoxS instead of RoxS-BsHfq as BsHfq was not expressed.

[Response] Thank you. We corrected the labeling accordingly in Figure 2b.

b. The x-axis for *Staphylococcus epidermidis* should be RFP and not EGFP.

[Response] Thank you. We corrected the labeling accordingly in Figure 2b.

Reviewer #3 (Remarks to the Author):

The authors have addressed the points I raised in my initial review. The side-by-side experimental comparison of BHR-sRNA and MicC-EcHfq strengthens the manuscript and will guide future users on which system to use depending on the host. The new results on the combination of CRISPRi and sRNA are a great addition to the manuscript and provide the community with a new strategy to reach higher silencing levels. I believe this system will be used by the community and I recommend the publication of this work.

[Response] Thank you very much for your generous comments.

Other changes:

In the previously submitted version of the manuscript, we miscounted the number of bacterial strains showing successful knockdown of target genes ($P < 0.017$). We thus corrected the number in the revised manuscript as follows (Revised manuscript page 9):

“The BHR-sRNA system allowed successful knockdown of the reporter genes tested (Supplementary Table 7) in 15 out of 16 bacteria tested, ...”